# Synthesis and Biological Evaluation of Diversified Hamigeran B Analogs as Neuroinflammatory Inhibitors and Neurite Outgrowth Stimulators

**DOI:** 10.3390/md18060306

**Published:** 2020-06-11

**Authors:** Ruo-Xin Li, Rui Han, Guo-Jie Wu, Fu-She Han, Jin-Ming Gao

**Affiliations:** 1Shaanxi Province Key Laboratory of Natural Products & Chemical Biology, College of Chemistry & Pharmacy, Northwest A&F University, Yangling 712100, China; rxli2014@hotmail.com (R.-X.L.); hanrui0820@nwsuaf.edu.cn (R.H.); 2Jilin Province Key Laboratory of Green Chemistry and Process, Changchun Institute of Applied Chemistry, Chinese Academy of Sciences, 5625 Renmin Street, Changchun 130022, China; gjwu@ciac.ac.cn

**Keywords:** nitric oxide inhibitor, neurotrophic activity, norditerpenoid analogues, anti-neuroinflammatory activity, structure-activity relationship

## Abstract

We describe the efficient synthesis of a series of new simplified hamigeran B and 1-hydroxy-9-epi-hamigeran B norditerpenoid analogs (23 new members in all), structurally related to cyathane diterpenoid scaffold, and their anti-neuroinflammatory and neurite outgrowth-stimulating (neurotrophic) activity. Compounds **9a**, **9h**, **9o**, and **9q** exhibited moderate nerve growth factor (NGF)-mediated neurite-outgrowth promoting effects in PC-12 cells at the concentration of 20 μm. Compounds **9b**, **9c**, **9o**, **9q,** and **9t** showed significant nitric oxide (NO) production inhibition in lipopolysaccharide (LPS)-activated BV-2 microglial cells, of which **9c** and **9q** were the most potent inhibitors, with IC_50_ values of 5.85 and 6.31 μm, respectively. Two derivatives **9q** and **9o** as bifunctional agents displayed good activities as NO production inhibitors and neurite outgrowth-inducers. Cytotoxicity experiments, H_2_O_2_-induced oxidative injury assay, and ELISA reaction speculated that compounds may inhibit the TNF-α pathway to achieve anti-inflammatory effects on nerve cells. Moreover, molecular docking studies provided a better understanding of the key structural features affecting the anti-neuroinflammatory activity and displayed significant binding interactions of some derivatives (like **9c**, **9q**) with the active site of iNOS protein. The structure-activity relationships (SARs) were also discussed. These results demonstrated that this structural class compounds offered an opportunity for the development of a new class of NO inhibitors and NGF-like promotors.

## 1. Introduction

In recent years, the prevalence of neurodegenerative disorders is increasing [1,2,3,4]. Dementias are responsible for the greatest burden of neurodegenerative diseases. For example, it is estimated that there are currently almost 47 million people worldwide living with dementia. Alzheimer’s disease (AD) is the most common type of dementia and it causes aberrant synapse pruning in neurologic disorders. Individuals from the aging population are suffering from AD [5,6,7,8], without any available disease-modifying treatments to prevent or treat cognitive deficits associated [9,10,11]. There has been an explosion of new findings in the nervous system, from contributions to migration of cells and synapse elimination during development to detrimental damage of nerve cells in autoimmunity and aberrant synapse pruning in neurologic disorders [12,13,14]. Meanwhile, studies suggest that the promotion of an anti-inflammatory response may slow or prevent diseases [15]. Therefore, there is a continuous need to search for high potent compounds as primary sources of medicines to optimize drug discovery and to develop more effective therapies. On the other hand, nerve growth factor (NGF), brain-derived neurotrophic factor (BDNF), neurotrophins like NT-3, NT-4, NT-5, have attracted attention as potential therapeutics for severe neurodegenerative diseases such as AD or as regeneration-promoting compounds [16]. However, their protein properties such as complex structures and high molecular weight eliminated them for using as clinical medication. Hence, the search for small molecules with the NGF biological functions or the NGF-induced neurotrophic activity arouses high interest [17].

The hamigeran diterpenoids, possessing a unique [5,6,6] or [5,6,7] fused tricyclic framework (Figure 1), also known as A-B-C rings, are structurally intriguing, marine-derived natural products. Among them, hamigerans A and B (**1** and **2**), discovered in 2000 from the poecilosclerid sponge, *Hamigera tarangaensis* [18], share the first naturally occurring [5,6,6] skeleton. Interestingly, the cyathanes comprise a structurally diverse class of diterpenoids with over 100 members reported from higher fungi thus far [19], featuring a unique [5,6,7] tricarbocyclic scaffold (Figure 1). Except that the seven-membered ring in cyathane scaffold is replaced by a benzene ring in hamigerans, they all have tricarbocyclic ring system, multiple stereogenic centers, and are punctuated by carbons at a variety of oxidation states. Very recently, our group discovered a series of novel natural cyathane diterpenoids with neurotrophic and anti-neuroinflammatory effects from higher basidiomycetes such as *Cyathus africanus*, *Sarcodon scabrosus* [20,21], such as sarcodonin G (**3**), striatoid B (**4**), cyafricanin C (**5**), and allocyathin B_2_ (**6**). The two small families of natural products have aroused considerable interest from the research communities of natural products, pharmacology, and synthetic chemistry because of their unique structures with intriguing biological potential [22,23,24,25]. Many synthetic endeavors have been devoted to the synthesis of hamigeran B (**2**), and our group achieved the total synthesis of hamigeran B in 2018 [23,24,25,26,27]. However, the novel pharmacological properties of hamigeran B analogs have not been assessed in anti-neuroinflammatory and neurotrophic activity so far. Because of the similarity of the ABC ring scaffold in their structures, we speculated the hamigeran B derivatives may also possess analogous neurological activity to the cyathanes. As a result, the substantial interest in the synthesis has resulted in the biological investigation on neurite-outgrowth stimulation and anti-neuroinflammatory activity. Herein, we present a concise synthesis of simplified hamigeran B and 1-hydroxy-9-*epi*-hamigeran B analogs and biological evaluation thereof.

## 2. Results

### 2.1. Chemistry

The synthetic route for the synthesis of the hamigeran B skeleton is illustrated in Scheme 1. The synthesis of diol **9a** commenced with preparation of chiral phenolic compound **7** starting from precursor **S1** which was prepared from a readily affordable 2-methyl-1,3-cyclopentanedione over six steps with 13% total yield, according to the previously reported protocol [26]. The compound **S1** was treated by OsO_4_ and NaIO_4_ to give the aldehyde **7** a 70% yield, which was unstable in air [26]. The tricyclic compound **8** is envisioned as being constructed through a key intramolecular Friedel–Crafts cyclization of aldehyde **7** with 70% yield. The exclusively high regioselectivity may be attributed to the formation of a coordination intermediate, which fixes the nucleophilic site at the ortho position to the phenolic OH. The compound **8** was treated with PCC to give the oxidative product **9b** in 70% yield. After obtaining the dehydrated intermediate **10**, the regioselective dihydroxylation of the double bond in the central B ring of **10** with OsO_4_ and 4-methylmorpholine N-oxide (NMO) proceeded anti to the angular methyl group to give the diol **9a** in 60% yield.

The synthetic route for the synthesis of 1-hydroxyl-9-*epi*-hamigeran B intermediate is illustrated in Scheme 2. The synthesis of intermediate diol **9p**, began with the known chiral cyclopentaneone **11** as the previous study [28]. Using imidazole and TBSCl to treat compound **11** afforded the α-hydroxy protected compound **12**. Isopropylation of the compound **12** with i-PrI followed by a base enolization with LiHMDS and Comins’ reagent (PhNTf_2_ is comins’ reagent, named after a person, Daniel Comins) provided enol triflate **13** in two steps with 70% yield. For the coupled product **14**, we preferred to examine the Suzuki cross-coupling of **13** with pinacol boronate instead of the corresponding boronic acid that was used previously in Trost’s synthesis [29]. Oxidation of the olefinic **14** was conducted by using a combination of OsO_4_ and NaIO_4_ to give 90% yield of the aldehyde **15**, which was unstable in the air [26]. The compound **16** was envisioned as being constructed via an intramolecular Friedel–Crafts cyclization as the key reaction of **15** followed by protection of the phenol hydroxyl with MeI. We are failed to obtain the pure **16** as the by-product has the similar polarity. To our delight, the by-product can be removed next step. The compound **16** was affected by PCC to give the oxidative product **9o**. A couple of enantiomers with ipsilateral dihydroxyl obtained by regioselective dihydroxylation of TBS protected **17** are difficult to purify. Therefore, the compound **17** were constructed through removing the protective group followed by dehydration. The regioselective dihydroxylation of the double bond in the B ring of **17** with OsO_4_ and NMO anti to the angular methyl group afforded the intermediate diol **9p**. We have established a concise route for the synthesis of 1-hydroxyl-9-epi-hamigeran B cores **9p** and **9o** that proceeds via an eight-step sequence from the known ketone **11**. To the best of our knowledge, this synthetic route represents the first time that can accomplish the synthesis of 1-hydroxyl-9-*epi*-hamigeran B intermediates within 10 steps from a known compound. Moreover, most of the transformations employed herein could be performed on multigram scale. This makes our synthetic route highly useful for mass production.

The hamigeran B skeleton-based derivatives **9b**–**n** and 1-hydroxyl-9-*epi*-hamigeran B-based **9o**-**w** obtained (Figure 2) were prepared according to the methods in the section of Materials and Methods. Protecting the hydroxyl of **9a** afforded the corresponding compound **9h**-**n**. Hydrogenation of **9h** and **9g** proceeded very efficiently to provide the hydrogenated product **9c** and **9e**. The compounds **9h**, **9m,** and **9n** were constructed via esterification from **9a**. Hydrolysis and concomitant oxidation proceeded **9c** to produce **9d**, followed by the condensation which afforded **9g**. Using trifluoroacetic acid to treat compound **9e** afforded the product **9f**. Compounds **9q** and **9s** were constructed via removing methyl on the phenolic group of **9o** and **9r**. The compound **9t** was established through simply eliminating 1-Hydroxy of **9s**. Protecting the hydroxyl of **9p** afforded the corresponding compound **9u**-**w**. There are also six annellation compounds (**9e**, **9g**, **9i**–**j**, **9l**, **9w**) and two trisubstituted ester molecules (**9m** bearing Boc-alaester and **9n** bearing nitrobenzoate). After chromatographic purification, all compounds including **9a**–**w** were subjected to spectroscopic analysis, and their structures and purity (≥95%) for biological analysis were confirmed by means of HRESIMS and ^1^H and ^13^C NMR analysis.

### 2.2. Biological Evaluation

According to previously reported methods [30,31], we used rat pheochromocytoma PC12 cells as a model system of neuronal differentiation to measure the effects of the synthesized compounds hamigeran B (**2**), **9a**–**n** and **9o**–**w** on neurite outgrowth, with NGF (10 ng/mL) as positive control. Specifically, as shown in Figure 3, most of them in combination with NGF (10 ng/mL) showed NGF-mediated neurite-outgrowth promoting activities, and **9a**, **9h**, **9o**, and **9q** exhibited significant neuritogenic effects with the percentage of neurite-bearing PC12 cells of 15.94%, 14.99%, 24.23%, and 16.68%, respectively, compared to NGF control cells (9.35%) and the parent molecule **2** (11.34%). Among them, **9o**, possessing an ether bond, and in particular large bulky TBS and OMe groups, was the most potent neurotrophic NGF-like inducer.

The inhibitory effects of the hamigeran B derivatives (**9a**–**n**) and 1-hydroxyl-9-epi-hamigeran B ones (**9o**–**w**) on LPS-stimulated NO production in BV2 cells were assessed according to our reported methods [32,33]. The results of inhibitory effects are depicted in Figure 4. As a result, most of the tested compounds showed inhibitory effects on NO production, and **9b**–**c**, **9f**, **9o**, **9q,** and **9t** exerted significant effects, with IC50 values in the approximate range 5.8–24 μm. Among them, the most potent inhibitors, **9c** (IC_50_ = 5.85 μm) and **9q** (IC_50_ = 6.31 μm) showed a comparable inhibition potency to natural product quercetin (IC_50_ = 4.3 μm). The most important thing is that they can be cheaply stored and transported as the less deliquescent than quercetin.

The effect on NO production could technically be due to toxicity on the cells. To exclude the possibility that their inhibitory activity was simply due to the cytotoxicity of the tested compounds, a cytotoxicity assay was conducted in BV-2 cells. As shown in Figure 5, when compared to the vehicle control at 10 μm, all 16 compounds failed to affect cell viability significantly. While the compounds display mild toxicity, the toxicity cannot account for the other observed effects. It is further proved that 16 compounds have certain anti-neuritis activity, which has nothing to do with cytotoxic activity.

The accumulation of H_2_O_2_ can lead to amyloid production, dopamine oxidation, and cerebral ischemia. Therefore, the formed H_2_O_2_ can be easily converted into highly toxic hydroxyl radicals generated by Fenton’s chemical method to damage lipids, proteins, and DNA. The oxidative damage may lead to mitochondrial dysfunction, calcium imbalance, inflammation, and apoptosis of nerve cells [34]. At a concentration of 10 μm, compound **9o** has better protective activity in the oxidative stress response of PC12 cells caused by H_2_O_2_ and other compounds have weak activity in Figure 6. 

TNF-α is a cytokine produced in response to infection and is closely related to the NF-κB pathway in the inflammatory response [35]. Based on the above experiments, in order to further prove the anti-neuritis activity of **9b**, **9c**, **9o,** and **9q**, we used enzyme-linked immune response (ELISA) to determine the secretion of TNF-α in BV-2 cells stimulated by LPS under the action of the above compounds. When BV-2 were treated with different concentrations of **9b**, **9c**, **9o,** and **9q**, TNF-α was reduced to varying degrees in a dose-dependent manner in Figure 7.

## 3. Discussion

Based on the NO inhibition profile of these hamigeran B analogs (Figure 4), the structure-activity relationships (SARs) are summarized in Figure 8: (1) For whether hamigeran B or 9-*epi*-hamigeran B series, the [5,6,6]-tricarbocyclic system (ABC ring) is essential to activity, and introduction of an *N*-or *O*-heterocyclic ring (D ring) at C-7/C-8, or acylation or ether bond of three hydroxyl groups in **9a** were detrimental to or led to a loss in activity (**9e**, **9g**, **9k**, **9w**); (2) for hamigeran B series, hydrogenation of the double bond at C-3/C-4 is crucial to improved activity (**9c** IC_50_ = 5.85 μm) relative to **9h**, hydrogenation of the keto group at C-8 enhanced activity (**9f**, IC_50_ = 24.91 μm vs. **9d**, IC_50_ > 50 μm); for 9-*epi*-hamigeran B series, the presence of the phenol hydroxyl at C14 was important for activity (**9q**, IC_50_ = 6.31 μm vs. **9o**, IC_50_ = 11.9 μm, the IC_50_ concentration curves in Appendix A), and introduction of a large steric hindrance group (a sterically hindered group TBS), that is, TBS, at C-1 was able to highly increase activity (**9o)** compared to **9r**; the large conjugated system with benzene-ring was important for enhancing the activity, for example, **9t** (IC_50_ = 11.98 μm) with a conjugated cyclpentan-1,3-diene unit was far more active in comparison with **9s**.

Cytotoxicity experiments showed that 16 compounds had no cytotoxic activity, proving that anti-neuritis activity was not caused by cytotoxicity. Therefore, compounds **9b**, **9c**, **9o,** and **9q** have no neuroprotective activity under the action of oxidative stress, further indicating that the 4 compounds have good anti-neuritis activity. Using ELISA reaction, it was found that **9b**, **9c**, **9o,** and **9q** and 4 compounds all inhibited the secretion of TNF-α in LPS activated BV-2 cells. Among them, the inhibitory effect of **9b** was particularly obvious, and the tumor necrosis factor (TNF-α) was down-regulated in a dose-dependent manner. TNF-α is an important factor in the NF-κB inflammation pathway, and it is speculated that compounds may inhibit the pathway to achieve anti-inflammatory effects on nerve cells.

In order to investigate the putative binding mode of the potent inhibitors **9b**, **9c**, **9o,** and **9q** on the inducible nitric oxide synthase (iNOS), a molecular modeling study was carried out. As a result, the binding mode predicted for **9b**, **9c**, **9o,** and **9q** in the docking studies is shown in Figure 9, the phenolic hydroxyl group at the C-14 position in **9b** facilitated hydrogen bonding interaction with TYR-373 (binding-length: 2.124 Å) and ASP-382 (binding-length: 2.079 Å and 2.325 Å). The carboxide of acetate group at the C-8 position in **9c** was favorably placed in the pocket facilitating hydrogen bonding interaction with TRP-346 (binding-length: 3.304 Å) and TYR-373 (binding-length: 1.984 Å), and the carboxide of acetate group at C-7 position facilitated hydrogen bonding interaction with GLN-263 (binding-length: 1.948 Å). The carboxide at the C-7 position in **9o** was favorably placed in the pocket facilitating hydrogen bonding interaction with TYR-373 (binding-length: 3.548 Å). Intriguingly, the protected α-hydroxy with TBS in **9q** facilitated hydrogen bonding with GLN-263 (binding-length: 1.996 Å and 3.311 Å). The results showed that the four compounds are stabilized by different and significant binding interactions (i.e., hydrogen bonding). These interactions were determined between the synthesized compounds and the active site of the iNOS enzyme model; furthermore, the calculated binding-lengths were in good agreement with the experimental IC_50_ values. The two most active compounds, that is, **9c** and **9q**, in these series exhibited several strong interactions with the active site of the enzyme (Figure 9). 

## 4. Materials and Methods

### 4.1. Chemistry

#### 4.1.1. General

The NMR spectra (detail in Appendix A) were recorded on Bruker AV at 300 and 500 MHz. Chemical shifts are given in ppm relative to TMS or the appropriate solvent resonance as the internal standard (CDCl_3_: 7.26; 77.16 ppm). High resolution mass spectra were obtained from IonSpec 4.7 Tesla FTMS mass spectrometer (MALDI), Bruker APEXIII 7.0 TESLA FTMS (ESI). Thin layer chromatography was conducted on Merck 60 F254 pre-coated silica gel plates. Column chromatography (CC) was carried out by normal silica gel (40–60 µm, 200–400 mesh, Silicycle P60). All reagents were obtained from commercial sources and used without further purification. Non-aqueous reactions were conducted under an inert atmosphere of argon in flame-dried glassware. Anhydrous solvents were treated under argon atmosphere as follow: tetrahydrofuran and diethyl ether were distilled from sodium, dichloromethane and toluene were distilled from calcium hydride. Anhydrous acetonitrile (Adamas, SafeDry, with molecular sieves) was commercially available.

#### 4.1.2. Detailed Procedure for Synthesis of Compounds **S1**, **7**, **8**, **10**, **12**–**17** and **9a**–**w**

*(S)*-*3*-*(2*-*isopropyl*-*5*-*methyl*-*5*-*(3*-*methylbut*-*2*-*en*-*1*-*yl)cyclopent*-*1*-*en*-*1*-*yl)phenol* (**S1**). To a solution of (*S*)-2-isopropyl-5-methyl-5-(3-methylbut-2-en-1-yl)cyclopent-1-en-1-yl trifluoromethanesulfonate (2.1 g, 6.2 mmol) in the mixed solvents of DMF (60 mL) and EtOH (60 mL), (3-hydroxyphenyl)boronic acid (1.7 g, 12.4 mmol), palladacycle [36] (0.18 g, 0.3 mmol), K_2_CO_3_ (1.7 g, 12.4 mmol) were added at room temperature under nitrogen. The resulting mixture was stirred at the same temperature for 24 h. The reaction mixture was then poured into water (60 mL) and extracted with ethyl acetate (3 × 60 mL). The organic layers were combined, washed with brine (5 × 15 mL), dried over Na_2_SO_4_ and concentrated in vacuum. The residue was purified by silica-gel CC to afford the desired cross-coupling product **S1** (5.50 g, 90%) as a colorless oil (*R*_f_ = 0.48 petroleum ether/ethyl acetate = 5/1). ^1^H NMR (300 MHz, CDCl_3_) δ 6.78–6.74 (m, 1H), 6.66 (dt, *J* = 7.6, 1.2 Hz, 1H), 6.58 (dd, *J* = 2.6, 1.4 Hz, 1H), 5.23–5.17 (m, 2H), 4.98 (s, 1H), 2.46–2.37 (m, 1H), 2.32 (d, *J* = 14.6 Hz, 2H), 2.04–1.96 (m, 2H), 1.95–1.83 (m, 1H), 1.73 (d, *J* = 1.5 Hz, 3H), 1.69–1.57 (m, 4H), 1.02 (s, 3H), 0.95 (t, *J* = 7.1 Hz, 6H). ^13^C NMR (75 MHz, CDCl_3_) δ 154.79, 144.81, 141.73, 140.41, 132.26, 128.81, 122.40, 121.9, 116.51, 113.10, 51.84, 38.16, 35.90, 28.11, 27.71, 26.40, 26.13, 21.59, 21.40, 18.01. HRESIMS *m*/*z* Calcd for C_20_H_28_O ([M + H]^+^): 285.2213; Found: 285.2219.

*(S)*-*2*-*(2*-*(3*-*hydroxyphenyl)*-*3*-*isopropyl*-*1*-*methylcyclopent*-*2*-*en*-*1*-*yl)acetaldehyde* (**7**). To a solution of NaIO_4_ (2.75 g, 12.89 mmol), pyridine (1.03 mL, 12.89 mmol), and **S1** (0.92 g, 3.22 mmol) in dioxane/H_2_O (30 mL/10 mL) OsO_4_ (0.039 M in H_2_O, 2.48 mL, 0.09 mmol) was added dropwise at room temperature. After stirring for 16 h, the reaction mixture was quenched with saturated aq. Na_2_SO_3_ (40 mL) and stirred for 30 min. The aqueous layer was extracted with EtOAc (3 × 20 mL). The combined organic layer was washed successively with H_2_O (2 × 20 mL) and brine (20 mL), dried over Na_2_SO_4_, and concentrated in vacuum. The residue was purified by silica-gel CC to provide **7** (0.68 g, 82% yield) as a colorless oil (*R*_f_ = 0.4, petroleum ether/EtOAc = 4:1). 

*(S)*-*2*-*(2*-*(3*-*hydroxyphenyl)*-*3*-*isopropyl*-*1*-*methylcyclopent*-*2*-*en*-*1*-*yl)acetaldehyde* (**8**). To a solution of **7** (0.6 g, 2.32 mmol) in THF (23 mL), ethylmagnesium bromide (0.3 M in THF, 7.7 mL, 2.32 mmol) was added dropwise at −78 °C and stirred for 1 h. The reaction mixture was heated to room temperature stirred overnight, then quenched with saturated aq. NH_4_Cl (30 mL), and then extracted with EtOAc (3 × 20 mL). The combined organic layer was washed successively with H_2_O (2 × 20 mL) and brine (20 mL), dried over Na_2_SO_4_ and concentrated in vacuum. The residue was purified by silica-gel CC to give **8** (0.42 g, 70% yield) as a white solid (*R*_f_ = 0.3, petroleum ether/EtOAc = 4:1). ^1^H NMR (300 MHz, CDCl_3_) δ 7.17 (t, *J* = 8.0 Hz, 1H), 6.98 (d, *J* = 7.8 Hz, 1H), 6.78 (d, *J* = 8.1 Hz, 1H), 5.20 (dd, *J* = 10.8, 6.7 Hz, 1H), 3.21 (p, *J* = 6.8 Hz, 1H), 2.60–2.33 (m, 3H), 1.83–1.60 (m, 3H), 1.16 (d, *J* = 6.9 Hz, 3H), 1.02 (d, *J* = 7.8 Hz, 6H). HRESIMS *m*/*z* Calcd for C_17_H_22_O_2_ ([M + Na]^+^): 281.1512; Found: 281.1506.

*(S)*-*1*-*isopropyl*-*3a*-*methyl*-*3,3a*-*dihydro*-*2H*-*cyclopenta[a]naphthalen*-*6*-*ol* (**10**). To a solution of **7** (0.6 g, 2.32 mmol) in THF (23 mL), ethylmagnesium bromide (0.3 M in THF, 7.7 mL, 2.32 mmol) was added dropwise at −78 °C and stirred for 1 h. The reaction mixture was then heated at 85 °C for 12 h. The reaction mixture was cooled to room temperature, quenched with saturated aq. NH_4_Cl (30 mL), and extracted with EtOAc (3 × 20 mL). The combined organic layer was washed successively with H_2_O (2 × 20 mL) and brine (20 mL), dried over Na_2_SO_4_, and concentrated in vacuum. The residue was purified by silica-gel CC to give **10** (0.39 g, 70% yield) as a brown solid (*R*_f_ = 0.5, petroleum ether/EtOAc = 4:1). ^1^H NMR (300 MHz, CDCl_3_) δ 7.14–6.97 (m, 2H), 6.74–6.51 (m, 2H), 6.06 (d, *J* = 9.8 Hz, 1H), 3.21 (p, *J* = 6.7 Hz, 1H), 2.45–2.33 (m, 2H), 1.90 (t, *J* = 5.9 Hz, 2H), 1.20 (d, *J* = 6.7 Hz, 3H), 1.03 (d, *J* = 6.6 Hz, 3H), 0.94 (s, 3H). HRESIMS *m*/*z* Calcd for C_17_H_20_O ([M − H]^−^): 239.1430; Found: 239.1438.

*(2S,3S)*-*3*-*((tert*-*butyldimethylsilyl)oxy)*-*2*-*methyl*-*2*-*(3*-*methylbut*-*2*-*en*-*1*-*yl)cyclopentan*-*1*-*one* (**12**). To a solution of **11** (5.70 g, 31.27 mmol) in DMF (60 mL) imidazole (2.56 g, 37.52 mmol) and TBSCl (5.66 g, 37.52 mmol) were added. After stirring overnight at room temperature, the reaction mixture was poured into aq. NaHCO_3_ (50 mL) and extracted with EtOAc (3 × 120 mL). The combined organic layer was washed with brine (5 × 30 mL), dried over Na_2_SO_4_ and concentrated in vacuum. The resulting residue was purified by silica-gel CC to afford **12** (7.42 g, 80%) as a colorless oil. (*R*_f_ = 0.4, petroleum ether/EtOAc = 15/1). ^1^H NMR (300 MHz, CDCl_3_) δ 5.10 (t, *J* = 7.7 Hz, 1H), 4.04 (t, *J* = 5.1 Hz, 1H), 2.47–2.36 (m, 1H), 2.29–2.00 (m, 4H), 1.96–1.87 (m, 1H), 1.70 (d, *J* = 1.7 Hz, 3H), 1.60 (s, 3H), 0.90 (d, *J* = 10.9 Hz, 12H), 0.09 (d, *J* = 4.8 Hz, 6H). ^13^C NMR (75 MHz, CDCl_3_) δ 220.19, 133.34, 119.71, 77.88, 53.71, 34.19, 34.15, 34.12, 28.93, 28.21, 28.19, 25.88, 25.84, 25.63, 25.60, 19.28, 19.25, 17.87, 17.82, 17.72, 17.69, 17.67, −4.39, −4.42, −5.09, −5.12. HRESIMS *m*/*z* Calcd for C_17_H_32_O_2_Si ([M + Na]^+^): 319.2056; Found: 319.2064. 

*(4S,5S)*-*4*-*((tert*-*butyldimethylsilyl)oxy)*-*2*-*isopropyl*-*5*-*methyl*-*5*-*(3*-*methylbut*-*2*-*en*-*1*-*yl)cyclopent*-*1*-*en*-*1*-*yl trifluoromethanesulfonate* (**13**). To a stirred suspension of NaH (60% dispersion in m ineral oil, 5.7 g, 142.5 mmol) in THF (160 mL) a solution of **12** (8.45 g, 28.5 mmol) in THF (40 mL) was added dropwise at 0 °C, then 2-iodopropane (28.5 mL, 285.0 mmol) was added. After refluxed overnight, the reaction mixture was quenched at 0 °C by adding H_2_O. Then 2M HCl (80 mL) was added, and the mixture stirred at room temperature for 1.5 h. Then brine was added, and the layers were separated. The aqueous layer was extracted with EtOAc (3 × 100 mL), and the combined organic layer was washed with brine (50 mL), dried over Na_2_SO_4_ and concentrated under reduced pressure. The resulting crude residue was purified by silica-gel CC to afford intermediate (8.46 g, 88% yield) as a colorless oil. The intermediate (8.57 g, 25.31 mmol) was dissolved in dry THF (50 mL) and cooled to −78 °C. LiHMDS (1.0 M in THF, 32.9 mL, 32.90 mmol) was added and stirred for 1 h. Then PhNTf_2_ (11.75 g, 32.90 mmol) dissolved in dry THF (50 mL) was slowly added and the mixture was heated to room temperature stirring for 3 h. Then brine (50 mL) was added to the reaction mixture, and the layers were separated. The aqueous layer was extracted with EtOAc (3 × 50 mL), and the combined organic layer was dried over Na_2_SO_4_ and concentrated in vacuum. The residue was purified by flash chromatography to afford **13** (10.84 g, 91% yield) as a colorless oil (*R*_f_ = 0.4, petroleum ether/CH_2_Cl_2_ = 50/1). ^1^H NMR (300 MHz, CDCl_3_) δ 5.28–5.22 (m, 1H), 4.04 (t, *J* = 7.7 Hz, 1H), 2.83 (p, *J* = 6.8 Hz, 1H), 2.44 (dd, *J* = 15.1, 7.8 Hz, 1H), 2.24–2.05 (m, 3H), 1.68 (d, *J* = 1.3 Hz, 3H), 1.58 (d, *J* = 1.4 Hz, 3H), 1.11 (s, 3H), 1.04 (d, *J* = 6.8 Hz, 3H), 0.93 (d, *J* = 3.9 Hz, 12H), 0.10 (d, *J* = 2.2 Hz, 6H). ^13^C NMR (75 MHz, CDCl_3_) δ 143.10, 135.49, 133.14, 120.74, 77.20, 50.24, 34.64, 32.82, 25.99, 25.74, 25.41, 22.68, 20.28, 20.15, 17.97, 17.51, −4.44, −4.89. HRESIMS *m*/*z* Calcd for C_21_H_38_F_3_O_4_SSi ([M + H]^+^): 471.2207; Found: 471.2197.

*3*-*((4S,5R)*-*4*-*((tert*-*butyldimethylsilyl)oxy)*-*2*-*isopropyl*-*5*-*methyl*-*5*-*(3*-*methylbut*-*2*-*en*-*1*-*yl)cyclopent*-*1*-*en*-*1*-*yl)phenol* (**14**). To a solution of vinyltriflate **13** (6.60 g, 14.0 mmol) in the mixed solvents of DMF (60 mL) and EtOH (60 mL), arylboronic acid (3.86 g, 28.0 mmol), palladacycle [36] (0.40 g, 0.06 mmol), K_2_CO_3_ (3.87 g, 28.0 mmol) were added at room temperature under nitrogen. When the vinyl triflate had disappeared as monitored by TLC analysis, the reaction mixture was then poured into water (60 mL) and extracted with ethyl acetate (3 × 60 mL). The organic layer was combined, washed with brine (5 × 15 mL), dried over Na_2_SO_4_, and concentrated in vacuum. The residue was purified by silica-gel CC to afford the desired cross-coupling product **14** (5.50 g, 80%, *R*_f_ = 0.4, petroleum ether/EtOA c = 5/1). ^1^H NMR (300 MHz, CDCl_3_) δ 7.17 (t, *J* = 7.8 Hz, 1H), 6.75–6.69 (m, 1H), 6.64 (dt, *J* = 7.6, 1.2 Hz, 1H), 6.55 (dd, *J* = 2.7, 1.4 Hz, 1H), 5.33–5.23 (m, 1H), 4.80 (dd, *J* = 6.8, 2.8 Hz, 1H), 4.02 (t, *J* = 7.6 Hz, 1H), 2.49–2.38 (m, 2H), 2.33–2.18 (m, 2H), 1.94 (dd, *J* = 15.0, 5.8 Hz, 1H), 1.68–1.56 (m, 6H), 0.91 (d, *J* = 8.6 Hz, 18H), 0.09 (s, 6H). ^13^C NMR (75 MHz, CDCl_3_) δ 154.87, 142.02, 130.45, 128.86, 123.00, 122.13, 116.36, 113.32, 80.67, 53.55, 37.32, 33.21, 27.50, 26.10, 25.95, 25.40, 21.46, 21.31, 18.21, 18.16, −4.27, −4.74. HRESIMS *m*/*z* Calcd for C_26_H_42_O_2_Si ([M + Na]^+^): 437.2846; Found: 437.2840.

*2*-*((1R,5S)*-*5*-*((tert*-*butyldimethylsilyl)oxy)*-*2*-*(3*-*hydroxyphenyl)*-*3*-*isopropyl*-*1*-*methylcyclopent*-*2*-*en*-*1*-*yl)acetaldehyde* (**15**). To a solution of **14** (6.34 g, 15.5 mmol) in the mixed solvent of dioxane (125 mL) and H_2_O (25 mL), NaIO_4_ (16.58 g, 77.5 mmol), pyridine (3.74 g, 46.5 mmol), and 15.5 mL solution of OsO_4_ in H_2_O (0.04 M) were added. The mixture was stirred at 80 °C until compound **14** had disappeared as monitored by TLC. H_2_O (100 mL) and EtOAc (200 mL) were added to the mixture. The organic phase was separated, and the aqueous layer was extracted with EtOAc (3 × 80 mL). The combined organic layer was dried over Na_2_SO_4_ and concentrated in vacuum. The resulting residue was purified by flash column chromatography to provide crude **15** (4.35 g, 80%) as a brown liquid (*R*_f_ = 0.5, petroleum ether/EtOAc = 4:1). ^1^H NMR (300 MHz, CDCl_3_) δ 7.54 (s, 1H), 7.17 (t, *J* = 7.8 Hz, 1H), 6.90–6.70 (m, 2H), 4.97 (s, 1H), 4.87 (d, *J* = 9.5 Hz, 1H), 4.22 (t, *J* = 8.4 Hz, 1H), 2.60–2.35 (m, 2H), 2.28 (dd, *J* = 15.4, 8.5 Hz, 1H), 2.10 (d, *J* = 14.6 Hz, 1H), 1.69 (d, *J* = 7.9 Hz, 1H), 1.04 (s, 3H), 0.96 (s, 9H), 0.91 (d, *J* = 7.7 Hz, 6H), 0.14 (d, *J* = 3.8 Hz, 6H). 

*(3S,3aR)*-*3*-*((tert*-*butyldimethylsilyl)oxy)*-*1*-*isopropyl*-*6*-*methoxy*-*3a*-*methyl*-*3,3a,4,5*-*tetrahydro*-*2H*-*cyclopenta[a]naphthalen*-*5*-*ol* (**16**). To a solution of aldehyde **15** (4.8 g) in THF (120 mL), ethylmagnesium bromide (0.5 M in THF, 27 mmol) was added dropwise at −78 °C and stirred for 1 h. After heating at 50 °C for 12 h, the reaction mixture was cooled to room temperature, quenched with saturated aq. NH_4_Cl (30 mL), and then extracted with EtOAc (3 × 80 mL). The combined organic layer was washed successively with H_2_O (150 mL) and brine (150 mL), dried over Na_2_SO_4_, and concentrated in vacuum. The resulting residue was purified by silica-gel CC to provide **16** (3.5 g, 70% yield) as a brown solid (*R*_f_ = 0.5, petroleum ether/EtOAc = 5:1). HRESIMS *m*/*z* Calcd for C_24_H_38_O_3_Si ([M + Na]^+^): 425.2482; Found: 425.2470.

*(3S,3aR)*-*1*-*isopropyl*-*6*-*methoxy*-*3a*-*methyl*-*3,3a*-*dihydro*-*2H*-*cyclopenta[a]naphthalen*-*3*-*ol* (**17**). To a solution of compound **16** (201 mg, 0.5 mmol) in THF (5 mL), TBAF (0.5 mL, 1M in THF) was added dropwise at 0 °C and stirred for 2 h. The reaction mixture was heated to room temperature, quenched with ammonium chloride aqueous (10 mL), and then extracted with EtOAc (3 × 10 mL). The combined organic layer was washed successively with brine (20 mL), dried over Na_2_SO_4_, and concentrated in vacuum. The resulting residue was purified by silica-gel CC to provide an intermediate (110 mg, 76% yield) as a white solid (*R*_f_ = 0.3, petroleum ether/EtOAc = 1:1). To a solution of intermediate (77 mg, 0.25 mmol) in THF (3 mL), *p*-toluenesulfonic acid (2.2 mg, 0.01 mmol) was added at room temperature. The reaction mixture was stirred at 75 °C for 8 h, quenched with aq. NH_4_Cl (5 mL), and then extracted with EtOAc (3 × 5 mL). The combined organic layer was washed successively with brine (10 mL), dried over Na_2_SO_4_, and concentrated in vacuum. The resulting residue was purified by silica-gel CC to provide **17** (62 mg, 92% yield) as a white solid (*R*_f_ = 0.7, petroleum ether/EtOAc = 1:1). ^1^H NMR (300 MHz, CDCl_3_) δ 7.19 (t, *J* = 8.0 Hz, 1H), 7.05 (d, *J* = 7.8 Hz, 1H), 6.89 (d, *J* = 10.0 Hz, 1H), 6.78 (d, *J* = 8.2 Hz, 1H), 6.15 (d, *J* = 9.9 Hz, 1H), 4.14 (d, *J* = 4.5 Hz, 1H), 3.85 (s, 3H), 3.30 (h, *J* = 6.9 Hz, 1H), 2.81 (dd, *J* = 17.0, 4.4 Hz, 1H), 2.45 (d, *J* = 17.0 Hz, 1H), 1.20 (d, *J* = 6.9 Hz, 3H), 1.09 (d, *J* = 6.9 Hz, 3H), 1.02 (s, 3H). ^13^C NMR (75 MHz, CDCl_3_) δ 155.22, 145.29, 133.38, 132.56, 131.19, 127.52, 120.63, 119.63, 109.50, 79.42, 55.61, 55.31, 38.84, 26.92, 22.12, 21.62, 21.27. HRESIMS *m*/*z* Calcd for C_18_H_22_O_2_ ([M + Na]^+^): 293.1512; Found: 293.1509.

*(3aR,4R,5S)*-*1*-*isopropyl*-*3a*-*methyl*-*3,3a,4,5*-*tetrahydro*-*2H*-*cyclopenta[a]naphthalene*-*4,5,6*-*triol* (**9a**). To a solution of 4-methylmorpholine N-oxide (0.78 g, 5.80 mmol) and **10** (0.35 g, 1.45 mmol) in THF/H_2_O (12 mL/3 mL), osmium tetroxide (0.039 M in H_2_O, 1.86 mL, 0.072 mmol) was added dropwise at room temperature. The reaction mixture was stirred at room temperature for 12 h. The reaction mixture was then quenched with saturated solution of sodium sulfite (20 mL) and stirred for 30 min. The mixture was extracted with ethyl acetate (3 × 10 mL). The combined organic layer was washed successively with H_2_O (2 × 10 mL) and brine (10 mL), dried over Na_2_SO_4_ and concentrated under reduced pressure. The crude reaction mixture was purified by silica-gel CC to provide **9a** (0.24 g, 60% yield) as a white solid (*R*_f_ = 0.3, petroleum ether/ethyl acetate = 4:1). ^1^H NMR (300 MHz, Chloroform-*d*) δ 8.65 (s, 1H), 7.19 (t, *J* = 7.9 Hz, 1H), 6.97 (d, *J* = 7.7 Hz, 1H), 6.87–6.77 (m, 1H), 5.03 (dd, *J* = 11.0, 4.0 Hz, 1H), 3.77–3.58 (m, 2H), 3.27 (p, *J* = 6.9 Hz, 1H), 2.63–2.50 (m, 2H), 2.19 (dt, *J* = 13.4, 9.0 Hz, 1H), 1.61–1.70 (m, 1H), 1.44 (d, *J* = 11.0 Hz, 1H), 1.20 (d, *J* = 6.9 Hz, 3H), 1.10–1.00 (m, 6H). ^13^C NMR (75 MHz, CDCl_3_) δ 158.57, 149.47, 131.85, 129.09, 128.94, 119.86, 119.06, 115.83, 77.45, 77.02, 76.60, 75.52, 70.18, 54.68, 31.15, 29.83, 27.51, 22.29, 22.04, 21.19. HMRS Calcd for C_17_H_22_O_3_ ([M + Na]^+^): 297.1461; Found: 297.1461.

*(S)*-*6*-*hydroxy*-*1*-*isopropyl*-*3a*-*methyl*-*2,3,3a,4*-*tetrahydro*-*5H*-*cyclopenta[a]naphthalen*-*5*-*one* (**9b**). The compound **8** (26 mg, 0.1 mmol) in dichloromethane (0.5 mL) was added to a stirred solution of PCC (86 mg, 0.4 mmol), NaOAc (33 mg, 0.4 mmol), and celite (20 mg) in dichloromethane (1 mL) at 0 °C. Then the solution was stirred at room temperature for 2 h. The reaction mixture was filtered through celite and washed repeatedly with Et_2_O. The solvent was removed under reduced pressure. The crude reaction mixture was purified by silica-gel CC to provide the ketone **9b** (18 mg, 70% yield) as a white solid (*R*_f_ = 0.6, petroleum ether/ethyl acetate = 4:1). ^1^H NMR (300 MHz, CDCl_3_) δ 12.71 (s, 1H), 7.45 (t, *J* = 8.0 Hz, 1H), 6.96 (d, *J* = 7.6 Hz, 1H), 6.85 (d, *J* = 8.4 Hz, 1H), 3.28–3.14 (m, 1H), 2.74 (s, 2H), 2.60 (dd, *J* = 8.8, 6.5 Hz, 2H), 1.94–1.78 (m, 2H), 1.22 (d, *J* = 6.9 Hz, 3H), 1.13–1.02 (m, 6H). ^13^C NMR (75 MHz, CDCl_3_) δ 205.59, 162.91, 146.94, 138.43, 136.21, 133.21, 118.25, 116.15, 77.45, 77.02, 76.60, 54.45, 49.99, 36.70, 29.57, 27.40, 24.56, 21.42, 21.18. HRMS Calcd for C_17_H_20_O_2_ ([M + H]^+^): 257.1536; Found: 257.1532.

*(1R,3aR,4R,5S,9bR)*-*1*-*isopropyl*-*3a*-*methyl*-*2,3,3a,4,5,9b*-*hexahydro*-*1H*-*cyclopenta[a]naphthalene*-*4,5,6*-*triyl triacetate* (**9c**). To a solution of compound **9h** (100 mg, 0.25 mmol) in EtOH (5 mL), Pd(OH)_2_/C (25 mg, 10%) was added in a high-pressure autoclave. The reaction mixture was hydrogenated for 24 h under 32 atm H_2_ at room temperature. Pd(OH)_2_/C was removed by filtration and the filtration was concentrated under reduced pressure without further purification to provide **9c** (100 mg, quant) as a white solid (*R*_f_ = 0.6, petroleum ether/ethyl acetate = 2:1). ^1^H NMR (500 MHz, CDCl_3_) δ 7.30 (t, *J* = 8.6 Hz, 1H), 7.12 (d, *J* = 7.9 Hz, 1H), 6.97 (d, *J* = 8.0 Hz, 1H), 6.34 (d, *J* = 3.7 Hz, 1H), 4.90 (d, *J* = 3.6 Hz, 1H), 3.19 (d, *J* = 7.1 Hz, 1H), 2.45–2.38 (m, 1H), 2.23 (t, *J* = 1.7 Hz, 3H), 2.04–1.89 (m, 7H), 1.96–1.88 (m, 1H), 1.66–1.57 (m, 2H), 1.46–1.35 (m, 2H), 1.19 (d, *J* = 1.4 Hz, 3H), 1.14 (dd, *J* = 6.2, 1.4 Hz, 3H), 0.71 (dd, *J* = 6.5, 1.4 Hz, 3H). ^13^C NMR (75 MHz, CDCl_3_) δ 170.65, 170.19, 169.44, 149.40, 139.26, 128.85, 127.78, 125.76, 120.02, 77.48, 77.06, 76.63, 75.02, 63.38, 53.61, 52.98, 42.59, 34.38, 30.95, 28.10, 26.10, 24.07, 21.44, 20.94, 20.87, 20.84. HRMS Calcd for C_23_H_30_O_6_ ([M + Na]^+^): 425.1935; Found: 425.1926.

*(1R,3aR,9bR)*-*6*-*hydroxy*-*1*-*isopropyl*-*3a*-*methyl*-*2,3,3a,9b*-*tetrahydro*-*1H*-*cyclopenta[a]naphthalene*-*4,5*-*dione* (**9d**). To a solution of lithium hydroxide (105 mg, 2.5 mmol) in H_2_O (2.5 mL), a solution of compound **9c** (100 mg, 0.25 mmol) in THF (2.5 mL) was added at 0 °C. The reaction mixture was stirred at room temperature for 12 h under oxygen atmosphere. To the mixture, 2N HCl (5 mL) was added and extracted with ethyl acetate (3 × 10 mL). The combined organic layer was washed successively with H_2_O (2 × 10 mL) and brine (10 mL), dried over Na_2_SO_4_, and concentrated under reduced pressure. The crude reaction mixture was purified by silica-gel CC to provide **9d** (48 mg, 70% yield) as a yellow solid (*R*_f_ = 0.5, petroleum ether/ethyl acetate = 4:1). ^1^H NMR (300 MHz, CDCl_3_) δ 11.91 (s, 1H), 7.56 (t, *J* = 8.0 Hz, 1H), 6.92 (dd, *J* = 13.0, 7.9 Hz, 2H), 3.49 (d, *J* = 9.0 Hz, 1H), 2.64 (dd, *J* = 9.1, 3.9 Hz, 1H), 2.38–2.24 (m, 1H), 1.92–1.52 (m, 3H), 1.33 (s, 3H), 1.22 (dt, *J* = 13.4, 6.5 Hz, 1H), 0.56 (d, *J* = 6.5 Hz, 3H), 0.42 (d, *J* = 6.5 Hz, 3H). ^13^C NMR (75 MHz, CDCl_3_) δ 199.69, 185.32, 164.45, 144.48, 138.52, 122.15, 118.58, 116.01, 77.45, 77.02, 76.60, 56.98, 56.58, 51.48, 33.82, 28.21, 26.98, 24.31, 23.15, 19.87. HRMS Calcd for C_17_H_20_O_3_ ([M + Na]^+^): 295.1305; Found: 295.1303.

*(3aR,3bR,6R,6aR,10bS)*-*6*-*isopropyl*-*2,2,3b*-*trimethyl*-*3a,3b,5,6,6a,10b*-*hexahydro*-*4H*-*cyclopenta [3,4]**naphtho[1*,*2*-*d][1,3]**dioxol*-*10*-*ol* (**9e**). To a solution of compound **9j** (31 mg, 0.1 mmol) in CH_2_Cl_2_ (1 mL), Pd(OH)_2_/C (25 mg, 10%) was added in a high-pressure autoclave. The reaction mixture was hydrogenated for 24 h under 30 atm H_2_ at room temperature. Pd(OH)_2_/C was removed by filtration and the filtration was concentrated under reduced pressure without further purification to provide **9e** (31 mg, quant) as a white solid (*R*_f_ = 0.7, petroleum ether/ethyl acetate = 4:1). ^1^H NMR (500 MHz, CDCl_3_) δ 7.42 (s, 1H), 7.08 (t, *J* = 7.9 Hz, 1H), 6.80 (dd, *J* = 7.8, 1.0 Hz, 1H), 6.72 (dt, *J* = 8.0, 0.9 Hz, 1H), 5.18 (d, *J* = 4.7 Hz, 1H), 4.14 (d, *J* = 4.8 Hz, 1H), 3.09 (d, *J* = 8.3 Hz, 1H), 2.24–2.14 (m, 2H), 2.05–1.98 (m, 1H), 1.94–1.87 (m, 1H), 1.72–1.64 (m, 1H), 1.49–1.43 (m, 4H), 1.25 (s, 3H), 1.12 (s, 3H), 0.66 (d, *J* = 6.7 Hz, 3H), 0.51 (d, *J* = 6.6 Hz, 3H). ^13^C NMR (75 MHz, CDCl_3_) δ 156.77, 137.22, 128.07, 123.15, 120.23, 113.39, 109.96, 79.80, 73.47, 51.96, 51.72, 41.45, 35.52, 28.90, 28.01, 27.37, 26.91, 26.44, 24.21, 21.38. HRMS Calcd for C_20_H_28_O_3_ ([M + Na]^+^): 339.1931; Found: 339.1927.

*(1R,3aR,4R,9bR)*-*4,6*-*dihydroxy*-*1*-*isopropyl*-*3a*-*methyl*-*1,2,3,3a,4,9b*-*hexahydro*-*5H*-*cyclopenta[a]naphthalen*-*5*-*one* (**9f**). To a solution of compound **9e** (32 mg, 0.1 mmol) in CH_2_Cl_2_ (1 mL), trifluoroacetic acid (35 μL, 0.4 mmol) was added at 0 °C. Then the reaction mixture was stirred at that temperature for 2 h and concentrated in vacuum. The crude reaction mixture was purified by silica-gel CC to provide **9f** (23 mg, 84% yield) as a white solid (*R*_f_ = 0.6, petroleum ether/ethyl acetate = 5:1) [37]. ^1^H NMR (300 MHz, CDCl_3_) δ 7.09 (t, *J* = 7.8 Hz, 1H), 6.84 (d, *J* = 7.7 Hz, 1H), 6.69 (d, *J* = 7.9 Hz, 1H), 5.55 (s, 1H), 3.74 (d, *J* = 21.4 Hz, 1H), 3.47–3.32 (m, 2H), 2.58–2.51 (m, 1H), 2.31–2.21 (m, 1H), 1.84–1.68 (m, 1H), 1.61–1.46 (m, 1H), 1.45–1.24 (m, 2H), 1.21 (s, 3H), 0.54 (dd, *J* = 17.7, 6.6 Hz, 6H). ^13^C NMR (75 MHz, CDCl_3_) δ 214.89, 152.69, 137.31, 126.78, 123.18, 120.46, 112.49, 77.45, 77.03, 76.61, 57.10, 54.33, 50.74, 36.64, 34.86, 28.23, 27.14, 24.83, 23.30, 19.58. HRMS Calcd for C_20_H_28_O_3_ ([M − H]^−^): 273.1485; Found: 273.1478.

*(3bR,6R,6aR)*-*6*-*isopropyl*-*2,2,3b*-*trimethyl*-*2,3b,4,5,6,6a*-*hexahydrocyclopenta[3,4]**naphtho[1,2*-*d]imidazol*-*10*-*ol* (**9g**). To a stirred solution of compound **9d** (27 mg, 0.1 mmol) and ammonium acetate (54 mg, 0.7 mmol) in AcOH (0.5 mL), acetone (8 μL, 0.11 mmol) was added. Then the reaction mixture was heated to 120 °C for 7 h. To the mixture was added ethyl acetate (5 mL) and washed with brine (10 mL) and sodium bicarbonate solution (10 mL), dried over Na_2_SO_4_ and concentrated under reduced pressure. The crude reaction mixture was purified by silica-gel CC to provide **9g** (22 mg, 71% yield) as a yellow solid (*R*_f_ = 0.5, petroleum ether/ethyl acetate = 4:1) [38]. ^1^H NMR (300 MHz, CDCl_3_) δ 10.88 (s, 1H), 7.35 (t, *J* = 7.9 Hz, 1H), 6.90 (dd, *J* = 17.9, 7.9 Hz, 2H), 3.32 (d, *J* = 9.7 Hz, 1H), 2.61–2.49 (m, 1H), 2.32–2.19 (m, 1H), 1.84 (q, *J* = 7.2 Hz, 2H), 1.55 (m, 7H), 1.29 (s, 3H), 1.06 (q, *J* = 6.7 Hz, 1H), 0.42 (dd, *J* = 8.4, 6.6 Hz, 6H). ^13^C NMR (75 MHz, CDCl_3_) δ 172.37, 160.54, 159.18, 141.56, 132.99, 122.18, 114.19, 113.24, 104.61, 77.44, 77.02, 76.60, 55.68, 51.96, 45.23, 37.53, 28.02, 27.82, 24.39, 24.02, 23.48, 19.66. HRMS Calcd for C_20_H_26_O_3_ ([M + H]^+^): 311.2118; Found: 311.2115.

*(3aR,4R,5S)*-*1*-*isopropyl*-*3a*-*methyl*-*3,3a,4,5*-*tetrahydro*-*2H*-*cyclopenta[a]naphthalene*-*4,5,6*-*triyl triacetate* (**9h**). To a solution of **9a** (0.1 g, 0.37 mmol) in pyridine (5 mL), acetic anhydride (0.5 mL, 5.5 mmol) was added at room temperature. The reaction mixture was heated at 100 °C for 3 h. The reaction mixture was cooled to room temperature. To the mixture, 2N HCl (5 mL) was added and stirred for 10 min. The mixture was extracted with ethyl acetate (3 × 10 mL). The combined organic layer was washed successively with H_2_O (2 × 10 mL) and brine (10 mL), dried over Na_2_SO_4_ and concentrated under reduced pressure. The crude reaction mixture was purified by silica-gel CC to provide **9h** (0.13 g, 86% yield) as a white solid (*R*_f_ = 0.5, petroleum ether/ethyl acetate = 2:1). ^1^H NMR (300 MHz, CDCl_3_) δ 7.37 (t, *J* = 7.9 Hz, 1H), 7.31–7.26 (m, 1H), 6.96 (dd, *J* = 8.0, 1.3 Hz, 1H), 5.18 (d, *J* = 4.6 Hz, 1H), 3.20 (p, *J* = 6.8 Hz, 1H), 2.62–2.45 (m, 2H), 2.26 (s, 3H), 2.02 (d, *J* = 11.4 Hz, 6H), 1.97–1.88 (m, 1H), 1.68–1.60 (m, 1H), 1.20 (d, *J* = 6.9 Hz, 3H), 1.09 (s, 3H), 1.02 (d, *J* = 6.7 Hz, 3H). ^13^C NMR (75 MHz, CDCl_3_) δ 170.93, 170.03, 169.26, 149.71, 146.22, 135.98, 131.65, 129.23, 125.08, 124.35, 121.15, 77.47, 77.04, 76.62, 75.32, 66.63, 52.04, 32.92, 29.68, 27.29, 23.46, 21.43, 21.25, 20.98, 20.91, 20.71. HRMS Calcd for C_23_H_28_O_6_ ([M + Na]^+^): 423.1778; Found: 423.1771.

*(3aR,3bR,10bS)*-*10*-*hydroxy*-*6*-*isopropyl*-*3b*-*methyl*-*3a,3b,5,10b*-*tetrahydro*-*4H*-*cyclopenta[3,4]**naphtho[1,2*-*d]**[1,3]**dioxol*-*2*-*one* (**9i**). To a stirred solution of compound **9a** (27 mg, 0.1 mmol) and pyridine (80 μL, 1 mmol) in CH_2_Cl_2_ (1.5 mL), a solution of triphosgene (45 mg, 0.15 mmol) in CH_2_Cl_2_ (0.5 mL) was added dropwise at −78 °C. Then the reaction mixture was allowed to warm to room temperature slowly. The reaction mixture was quenched using saturated aqueous NH_4_Cl followed by extraction of the aqueous phase with CH_2_Cl_2_. The organic extracts were washed with 1 M HCl, saturated NaHCO_3_, saturated NaCl, H_2_O, dried with MgSO_4_, and filtered. The crude reaction mixture was purified by silica-gel CC to provide **9i** (25 mg, 82% yield) as a white solid (*R*_f_ = 0.2, petroleum ether/ethyl acetate = 2:1) [39]. ^1^H NMR (300 MHz, CDCl_3_) δ 7.24 (t, *J* = 8.0 Hz, 1H), 6.91 (d, *J* = 7.7 Hz, 1H), 6.73 (d, *J* = 8.1 Hz, 1H), 6.06 (d, *J* = 7.7 Hz, 1H), 5.63 (s, 1H), 4.87 (d, *J* = 7.7 Hz, 1H), 3.21–3.00 (m, 1H), 2.58 (td, *J* = 9.9, 5.7 Hz, 2H), 2.39–2.24 (m, 1H), 1.79–1.69 (m, 1H), 1.19 (d, *J* = 6.8 Hz, 3H), 1.00 (d, *J* = 7.5 Hz, 6H). ^13^C NMR (75 MHz, CDCl_3_) δ 155.27, 148.98, 135.51, 130.76, 128.47, 119.97, 116.86, 114.50, 83.47, 77.45, 77.02, 76.60, 72.74, 51.98, 31.59, 29.96, 27.45, 23.09, 21.41, 21.34. HRMS Calcd for C_18_H_20_O_4_ ([M + Na]^+^): 323.1254; Found: 323.1250.

*(3aR,3bR,10bS)*-*6*-*isopropyl*-*2,2,3b*-*trimethyl*-*3a,3b,5,10b*-*tetrahydro*-*4H*-*cyclopenta**[3,4]**naphtho[1,2*-*d]**[1,3]**dioxol*-*10*-*ol* (**9j**). To a stirred solution of compound **9a** (55 mg, 0.2 mmol) and pyridinium p-toluenesulfonate acid (5 mg, 2% mmol) in CH_2_Cl_2_ (2 mL), 2,2-dimethoxypropane (52 mg, 0.5 mmol) was added at room temperature. Then the reaction mixture was stirred for 6 h at 40 °C. The solvent was removed under reduced pressure. The crude reaction mixture was purified by silica-gel CC to provide **9j** (51 mg, 82% yield) as a white solid (*R*_f_ = 0.5, petroleum ether/ethyl acetate = 4:1) [40]. ^1^H NMR (300 MHz, CDCl_3_) δ 7.19 (t, *J* = 7.9 Hz, 1H), 6.98 (d, *J* = 7.7 Hz, 1H), 6.81 (d, *J* = 8.0 Hz, 1H), 6.54 (s, 1H), 5.29 (d, *J* = 5.1 Hz, 1H), 4.25 (d, *J* = 5.1 Hz, 1H), 3.27–3.13 (m, 1H), 2.54 (t, *J* = 7.6 Hz, 2H), 2.31–2.17 (m, 1H), 1.69–1.56 (m, 1H), 1.44 (s, 3H), 1.19 (d, *J* = 6.8 Hz, 3H), 1.13 (s, 3H), 1.05–0.96 (m, 6H). ^13^C NMR (75 MHz, CDCl_3_) δ 156.16, 145.93, 133.34, 130.50, 128.71, 120.96, 119.54, 114.59, 110.18, 82.36, 73.51, 52.16, 31.80, 29.58, 27.83, 27.37, 26.98, 23.53, 21.62, 21.30. HRMS Calcd for C_20_H_26_O_3_ ([M + Na]^+^): 337.1774; Found: 337.1771.

*(3aR,4R,5S)*-*4,6*-*bis((tert*-*butyldimethylsilyl)oxy)*-*1*-*isopropyl*-*3a*-*methyl*-*3,3a,4,5*-*tetrahydro*-*2H*-*cyclopenta[a]naphthalen*-*5*-*ol* (**9k**). To a stirred solution of compound **9a** (27 mg, 0.1 mmol) and 2,6-lutidine (70 μL, 0.7 mmol) in CH_2_Cl_2_ (1 mL), TBSOtf (114 μL, 0.5 mmol) was added at −40 °C. Then the reaction mixture was stirred for 5 h at −40°C. The mixture was dilute with CH_2_Cl_2_ (10 mL) and washed with 1N HCl aqueous (4 mL) and brine (10 mL), dried over Na_2_SO_4_ and concentrated under reduced pressure. The crude reaction mixture was purified by silica-gel CC to provide **9k** (33 mg, 66% yield) as a white solid (*R*_f_ = 0.5, petroleum ether/ethyl acetate = 40:1) [41]. ^1^H NMR (300 MHz, CDCl_3_) δ 7.18 (t, *J* = 7.9 Hz, 1H), 7.08 (d, *J* = 7.8 Hz, 1H), 6.76 (d, *J* = 7.9 Hz, 1H), 4.72 (d, *J* = 2.0 Hz, 1H), 3.87 (s, 1H), 3.34 (d, *J* = 2.0 Hz, 1H), 3.22 (q, *J* = 6.8 Hz, 1H), 2.48 (dd, *J* = 8.8, 5.9 Hz, 2H), 2.15 (dt, *J* = 12.9, 7.7 Hz, 1H), 1.59–1.45 (m, 1H), 1.21 (d, *J* = 6.9 Hz, 3H), 1.08 (d, *J* = 7.8 Hz, 12H), 1.00 (d, *J* = 6.7 Hz, 3H), 0.84 (s, 9H), 0.39 (s, 3H), 0.25 (s, 3H), 0.15 (d, *J* = 2.8 Hz, 6H). ^13^C NMR (75 MHz, CDCl_3_) δ 154.47, 145.48, 134.50, 131.88, 127.52, 120.66, 116.07, 78.18, 71.17, 51.05, 32.14, 29.99, 27.38, 25.80, 25.72, 23.79, 21.66, 21.51, 18.02, 17.92, −3.35, −4.47, −4.74, −4.81. HRMS Calcd for C_29_H_50_O_3_Si_2_ ([M + Na]^+^): 525.3191; Found: 525.3185.

*(3aR,3bR,10bS)-6-isopropyl-2,2,3b-trimethyl-3a,3b,5,10b-tetrahydro-4H-cyclopenta[3,4]naphtho[1,2-d][1,3]dioxol-10-yl (tert-butoxycarbonyl)alaninate* (**9l**). To a stirred solution of compound **9j** (31 mg, 0.1 mmol) in CHCl_3_/THF (0.5/1.5 mL), Boc-l-Ala (68 mg, 0.36 mmol), DMAP (4 mg, 0.03 mmol), and EDCI (63 mg, 0.33 mmol) were added. The reaction was stirred for 18 h at room temperature. The solvent was removed in vacuum, and the crude reaction mixture was purified by silica-gel CC to provide **9l** (39 mg, 82% yield) as a white solid (R_f_ = 0.7, petroleum ether/ethyl acetate = 1:1). ^1^H NMR (300 MHz, CDCl_3_) δ 7.33 (t, J = 7.8 Hz, 1H), 7.23 (d, J = 7.7 Hz, 1H), 6.98 (d, J = 7.9 Hz, 1H), 5.36 (d, J = 8.1 Hz, 1H), 5.20 (d, J = 5.8 Hz, 1H), 4.73–4.61 (m, 1H), 4.24 (d, J = 5.9 Hz, 1H), 3.14 (p, J = 6.9 Hz, 1H), 2.53 (dt, J = 9.4, 5.1 Hz, 2H), 2.31–2.22 (m, 1H), 1.61–1.56 (m, 4H), 1.47 (d, J = 6.5 Hz, 9H), 1.36 (s, 3H), 1.20 (d, J = 6.9 Hz, 3H), 1.05 (s, 3H), 1.01–0.93 (m, 6H). ^13^C NMR (75 MHz, CDCl_3_) δ 155.29, 150.21, 146.67, 135.17, 130.46, 128.50, 127.07, 125.04, 120.85, 109.12, 82.52, 79.73, 72.03, 52.64, 49.56, 31.93, 29.99, 28.38, 27.42, 27.39, 26.39, 23.81, 21.48, 18.42. HRMS Calcd for C_28_H_39_NO_6_ ([M + Na]^+^): 508.2670; Found: 508.2661.

*(3aR,4R,5S)-1-isopropyl-3a-methyl-3,3a,4,5-tetrahydro-2H-cyclopenta[a]naphthalene-4,5,6-triyl tris(2-((tert-butoxycarbonyl)amino)propanoate)* (**9m**). To a stirred solution of compound **9a** (27 mg, 0.1 mmol) in CHCl_3_/THF (0.5/1.5 mL), Boc-l-Ala (68 mg, 0.36 mmol), DMAP (4 mg, 0.03 mmol), and EDCI (63 mg, 0.33 mmol) were added. The reaction was stirred for 18 h at room temperature. The solvent was removed in vacuum, and the crude reaction mixture was purified by silica-gel CC to provide **9m** (25 mg, 83% yield) as a white solid (R_f_ = 0.7, petroleum ether/ethyl acetate = 1:1) [42]. ^1^H NMR (300 MHz, CDCl_3_) δ 7.40 (t, J = 7.9 Hz, 1H), 7.06 (d, J = 8.1 Hz, 1H), 6.29 (d, J = 4.0 Hz, 1H), 5.68 (s, 1H), 5.17 (s, 2H), 4.45–4.18 (m, 3H), 3.23–3.07 (m, 1H), 2.50 (dt, J = 9.6, 5.2 Hz, 2H), 2.00 (d, J = 11.0 Hz, 1H), 1.64 (s, 3H), 1.55 (d, J = 7.2 Hz, 3H), 1.51–1.37 (m, 30H), 1.33 (d, J = 7.3 Hz, 3H), 1.18 (d, J = 6.8 Hz, 3H), 1.10–0.99 (m, 6H). ^13^C NMR (75 MHz, CDCl_3_) δ 172.83, 172.30, 171.83, 155.25, 154.90, 149.44, 145.53, 133.32, 130.01, 125.08, 124.17, 120.95, 79.73, 77.50, 77.28, 77.08, 76.65, 67.05, 50.23, 49.18, 33.41, 28.75, 28.35, 28.30, 26.88, 24.63, 21.26, 18.88, 18.14, 17.52. HRMS Calcd for C_41_H_61_N_3_O_12_ ([M + Na]^+^): 810.4147; Found: 810.4133.

*(3aR,4R,5S)*-*1*-*isopropyl*-*3a*-*methyl*-*3,3a,4,5*-*tetrahydro*-*2H*-*cyclopenta[a]naphthalene*-*4,5,6*-*triyltris(4*-*nitrobenzoate)* (**9n**). To a stirred solution of compound **9a** (55 mg, 0.2 mmol) and DMAP (54 mg, 0.8 mmol) in CH_2_Cl_2_ (1.5 mL), a solution of 4-nitrobenzoyl chloride (100 mg, 0.11 mmol) in CH_2_Cl_2_ (0.5 mL) was added. Then the reaction mixture was stirred for 3 h at room temperature. The mixture was dilute with CH_2_Cl_2_ (10 mL) and washed with brine (10 mL), dried over Na_2_SO_4_ and concentrated under reduced pressure. The crude reaction mixture was purified by silica-gel CC to provide **9n** (104 mg, 72% yield) as a white solid (*R*_f_ = 0.5, petroleum ether/ethyl acetate = 2:1) [43]. ^1^H NMR (300 MHz, CDCl_3_) δ 8.24–7.94 (m, 10H), 7.73–7.65 (m, 2H), 7.55 (d, *J* = 3.9 Hz, 2H), 7.15 (dd, *J* = 5.5, 3.8 Hz, 1H), 6.83 (d, *J* = 4.4 Hz, 1H), 5.60 (d, *J* = 4.4 Hz, 1H), 3.41 (p, *J* = 6.7 Hz, 1H), 2.74–2.53 (m, 2H), 2.06 (dt, *J* = 13.5, 9.0 Hz, 1H), 1.80 (dd, *J* = 12.7, 8.7, 3.4 Hz, 1H), 1.35–1.25 (m, 6H), 1.19 (d, *J* = 6.6 Hz, 3H). ^13^C NMR (75 MHz, CDCl_3_) δ 164.26, 163.32, 162.95, 150.75, 150.69, 150.42, 149.55, 147.01, 135.98, 134.85, 134.39, 133.89, 131.29, 131.12, 130.69, 130.35, 130.11, 125.92, 123.60, 123.39, 123.27, 123.18, 121.57, 77.48, 77.05, 76.63, 76.49, 68.51, 52.21, 32.70, 29.74, 27.39, 23.71, 21.54, 21.49. HRMS Calcd for C_38_H_31_N_3_O_12_ ([M + Na]^+^): 744.1800; Found: 744.1782.

*(3S,3aR)*-*3*-*((tert*-*butyldimethylsilyl)oxy)*-*1*-*isopropyl*-*6*-*methoxy*-*3a*-*methyl*-*2,3,3a,4*-*tetrahydro*-*5H*-*cyclopenta[a]naphthalen*-*5*-*one* (**9o**). The compound **16** (4.02 g, 10 mmol) in dichloromethane (10 mL) was added to a stirred solution of PCC (4.7 g, 22 mmol), NaOAc (1.5 g, 11 mmol) and celite (4 g) in dichloromethane (40 mL) at 0 °C. Then the solution was stirred at room temperature for 2 h. The reaction mixture was filtered through celite and washed repeatedly with Et_2_O. The solvent was removed under reduced pressure. The crude reaction mixture was purified by silica-gel CC to provide the ketone **9o** (2.9 g, 73% yield) as a colorless liquid (*R*_f_ = 0.3, petroleum ether/ethyl acetate = 5:1). ^1^H NMR (300 MHz, CDCl_3_) δ 7.43 (t, *J* = 8.1 Hz, 1H), 7.03 (d, *J* = 7.7 Hz, 1H), 6.87 (d, *J* = 8.4 Hz, 1H), 4.01 (dd, *J* = 6.5, 3.2 Hz, 1H), 3.91 (s, 3H), 3.25–3.00 (m, 2H), 2.85 (dd, *J* = 17.1, 6.4 Hz, 1H), 2.47–2.27 (m, 2H), 1.17 (d, *J* = 6.9 Hz, 3H), 1.07–0.98 (m, 6H), 0.89 (s, 9H), 0.07 (d, *J* = 2.0 Hz, 6H). ^13^C NMR (75 MHz, CDCl_3_) δ 199.08, 160.23, 143.06, 139.83, 133.73, 132.77, 120.49, 119.85, 110.52, 77.59, 77.51, 77.09, 76.66, 56.01, 53.96, 49.01, 40.07, 27.19, 25.82, 23.40, 21.22, 21.17, 18.09, −4.60, −4.89. HRMS Calcd for C_24_H_36_O_3_Si ([M + Na]^+^): 423.2326; Found: 423.2319.

*(3S,3aR,4S,5R)*-*1*-*isopropyl*-*6*-*methoxy*-*3a*-*methyl*-*3,3a,4,5*-*tetrahydro*-*2H*-*cyclopenta[a]naphthalene*-*3,4,5*-*triol* (**9p**). To a solution of 4-methylmorpholine N-oxide (68 mg, 0.5 mmol) and compound **17** (27 mg, 0.1 mmol) in THF/H_2_O (0.8 mL/0.2 mL), osmium tetroxide (0.039 M in H_2_O, 0.25 mL, 0.01 mmol) was added dropwise at room temperature. After stirring for 36 h, the reaction mixture was quenched with saturated solution of sodium sulfite (10 mL) and stirred for 30 min. The mixture was extracted with ethyl acetate (3 × 50 mL). The combined organic layer was washed successively with H_2_O (2 × 50 mL) and brine (10 mL), dried over Na_2_SO_4_, and concentrated under reduced pressure. The crude reaction mixture was purified by silica-gel CC to provide **9p** (18 mg, 60% yield) as a dark green solid (*R*_f_ = 0.3, petroleum ether/ethyl acetate = 1:1). ^1^H NMR (500 MHz, CDCl_3_) δ 7.25 (td, *J* = 7.9, 2.3 Hz, 1H), 6.97 (dd, *J* = 7.7, 2.3 Hz, 1H), 6.79 (dd, *J* = 8.4, 2.3 Hz, 1H), 5.16 (t, *J* = 3.0 Hz, 1H), 4.07 (dd, *J* = 5.5, 2.8 Hz, 1H), 3.93 (t, *J* = 3.0 Hz, 1H), 3.85 (d, *J* = 2.3 Hz, 3H), 3.12 (pd, *J* = 6.9, 2.2 Hz, 1H), 2.93–2.88 (m, 1H), 2.36 (dt, *J* = 17.3, 2.3 Hz, 1H), 1.13 (dd, *J* = 6.9, 2.4 Hz, 3H), 1.01–0.93 (m, 6H). ^13^C NMR (75 MHz, CDCl_3_) δ 157.44, 142.94, 134.76, 132.16, 128.70, 124.53, 120.24, 109.00, 78.64, 77.51, 77.08, 76.66, 76.20, 66.54, 55.53, 54.29, 40.59, 26.84, 23.14, 21.55, 21.14. HMRS Calcd for C_18_H_24_O_4_ ([M + Na]^+^): 327.1567; Found: 327.1563.

*(3S,3aR)*-*3*-*((tert*-*butyldimethylsilyl)oxy)*-*6*-*hydroxy*-*1*-*isopropyl*-*3a*-*methyl*-*2,3,3a,4*-*tetrahydro*-*5H*-*cyclopenta[a]naphthalen*-*5*-*one* (**9q**). To a solution of compound **9o** (40 mg, 0.1 mmol) in THF (1.5 mL), BBr_3_ (48 μL, 0.5 mmol) was added dropwise at −78 °C and stirred for 2 h. The reaction mixture was heated to room temperature slowly, quenched with ice water (5 mL), and then extracted with ethyl acetate (3 × 5 mL). The combined organic layer was washed successively with brine (10 mL), dried over Na_2_SO_4_ and concentrated under reduced pressure. The crude reaction mixture was purified by silica-gel CC to provide **9q** (29 mg, 76% yield) as a white solid (*R*_f_ = 0.2, petroleum ether/ethyl acetate = 2:1) [44]. ^1^H NMR (300 MHz, CDCl_3_) δ 12.76 (s, 1H), 7.45 (t, *J* = 8.0 Hz, 1H), 7.00 (d, *J* = 7.7 Hz, 1H), 6.86 (dd, *J* = 8.4, 1.0 Hz, 1H), 4.08 (dd, *J* = 6.3, 2.1 Hz, 1H), 3.33–3.21 (m, 2H), 3.02 (dd, *J* = 17.8, 6.2 Hz, 1H), 2.58–2.45 (m, 2H), 1.20 (d, *J* = 7.0 Hz, 3H), 1.13–1.06 (m, 6H), 0.89 (s, 9H), 0.07 (d, *J* = 2.0 Hz, 6H). ^13^C NMR (75 MHz, CDCl_3_) δ 205.99, 162.90, 144.34, 137.68, 136.29, 131.04, 118.33, 116.48, 114.92, 77.45, 77.03, 76.66, 76.61, 53.92, 45.88, 40.19, 27.32, 23.41, 21.39, 20.91. HRMS Calcd for C_23_H_34_O_3_Si ([M + Na]^+^): 409.2169; Found: 409.1616.

*(3S,3aR)*-*3*-*hydroxy*-*1*-*isopropyl*-*6*-*methoxy*-*3a*-*methyl*-*2,3,3a,4*-*tetrahydro*-*5H*-*cyclopenta[a]naphthalen*-*5*-*one* (**9r**). To a solution of compound **9o** (40 mg, 0.1 mmol) in THF (1 mL), TBAF (1M in THF, 0.15 mmol) was added dropwise at 0 °C and stirred for 1 h. The reaction mixture was cooled to room temperature, quenched with saturated solution of ammonium chloride (5 mL), and then extracted with ethyl acetate (3 × 5 mL). The combined organic layer was washed successively with H_2_O (10 mL) and brine (10 mL), dried over Na_2_SO_4_, and concentrated under reduced pressure. The crude reaction mixture was purified by silica-gel CC to provide **9r** (22 mg, 76% yield) as a white solid (*R*_f_ = 0.1, petroleum ether/ethyl acetate = 2:1). ^1^H NMR (300 MHz, CDCl_3_) δ 7.45 (t, *J* = 8.1 Hz, 1H), 7.06 (d, *J* = 7.8 Hz, 1H), 6.89 (d, *J* = 8.4 Hz, 1H), 4.08 (dd, *J* = 6.6, 2.8 Hz, 1H), 3.91 (s, 3H), 3.24–2.91 (m, 3H), 2.52–2.41 (m, 2H), 2.16 (s, 1H), 1.18 (d, *J* = 6.9 Hz, 3H), 1.10–1.01 (m, 6H). ^13^C NMR (75 MHz, CDCl_3_) δ 198.54, 160.21, 143.18, 139.50, 133.92, 132.53, 120.38, 119.88, 110.67, 77.49, 77.12, 77.07, 76.65, 56.00, 53.58, 48.11, 39.78, 27.22, 23.52, 21.32, 21.10. HRMS Calcd for C_18_H_22_O_3_ ([M + Na]^+^): 309.1461; Found: 309.1458.

*(3S,3aR)*-*3,6*-*dihydroxy*-*1*-*isopropyl*-*3a*-*methyl*-*2,3,3a,4*-*tetrahydro*-*5H*-*cyclopenta[a]naphthalen*-*5*-*one* (**9s**). To a solution of compound **9r** (43 mg, 0.15 mmol) in THF (1.5 mL), BBr_3_ (58 μL, 0.6 mmol) was added dropwise at −78 °C and stirred for 2 h. The reaction mixture was heated to room temperature slowly, quenched with ice water (5 mL), and then extracted with ethyl acetate (3 × 5 mL). The combined organic layer was washed successively with brine (10 mL), dried over Na_2_SO_4_, and concentrated under reduced pressure. The crude reaction mixture was purified by silica-gel CC to provide **9s** (31 mg, 76% yield) as a white solid (*R*_f_ = 0.6, petroleum ether/ethyl acetate = 1:1). ^1^H NMR (300 MHz, CDCl_3_) δ 12.75 (s, 1H), 7.43 (t, *J* = 8.0 Hz, 1H), 6.98 (d, *J* = 7.5 Hz, 1H), 6.84 (dd, *J* = 8.4, 1.0 Hz, 1H), 4.05 (s, 1H), 3.30–3.20 (m, 2H), 3.05–2.96 (m, 1H), 2.56–2.45 (m, 2H), 1.19 (d, *J* = 6.9 Hz, 3H), 1.09 (dd, *J* = 3.7, 2.9 Hz, 6H). ^13^C NMR (75 MHz, CDCl_3_) δ 205.98, 162.90, 144.34, 137.68, 136.30, 131.04, 118.33, 116.49, 114.92, 77.46, 77.03, 76.66, 76.61, 53.92, 45.88, 40.19, 27.32, 23.41, 21.39, 20.91. HRMS Calcd for C_17_H_20_O_3_ ([M + Na]^+^): 273.1485; Found: 273.1486.

*(S)*-*1*-*isopropyl*-*6*-*methoxy*-*3a*-*methyl*-*3a,4*-*dihydro*-*5H*-*cyclopenta[a]naphthalen*-*5*-*one* (**9t**). To a stirred solution of compound **9r** (31 mg, 0.11 mmol) in THF (1 mL) containing triphenylphophine (58 mg, 0.22 mmol) and *p*-nitrobenzoic acid (37mg, 0.22 mmol), DEAD (3 μL, 0.22 mmol) was added dropwise over 10 min period at 0 °C under argon, and the reaction mixture was stirred at 50 °C for 12 h. The solvent was removed under reduced pressure to leave an oil, which was purified by column chromatography to provide **9t** (21 mg, 72% yield) as a white solid (*R*_f_ = 0.4, petroleum ether/ethyl acetate = 2:1). ^1^H NMR (300 MHz, CDCl_3_) δ 7.52 (t, *J* = 8.1 Hz, 1H), 7.06 (dd, *J* = 7.8, 1.0 Hz, 1H), 6.89 (dd, *J* = 8.5, 1.0 Hz, 1H), 6.60 (d, *J* = 5.3 Hz, 1H), 6.53 (d, *J* = 5.3 Hz, 1H), 3.96 (s, 3H), 3.24 (p, *J* = 6.8 Hz, 1H), 2.92 (d, *J* = 15.3 Hz, 1H), 2.15 (d, *J* = 15.2 Hz, 1H), 1.33 (d, *J* = 6.9 Hz, 3H), 1.17–1.08 (m, 6H). ^13^C NMR (75 MHz, CDCl_3_) δ 197.41, 160.69, 147.11, 146.54, 139.83, 139.39, 134.27, 130.63, 119.20, 118.86, 109.95, 77.47, 77.04, 76.62, 56.52, 56.01, 50.95, 26.28, 22.54, 22.09, 19.09. HRMS Calcd for C_18_H_20_O_2_ ([M + H]^+^): 269.1536; Found: 269.1539.

*(3S,3aR,4S,5R)*-*1*-*isopropyl*-*6*-*methoxy*-*3a*-*methyl*-*3,3a,4,5*-*tetrahydro*-*2H*-*cyclopenta[a]naphthalene*-*3,4,5*-*triyl triacetate* (**9u**). To a solution of **9p** (30 g, 0.1 mmol) in pyridine (1.5 mL), acetic anhydride (0.13 mL, 1.5 mmol) was added at room temperature. The reaction mixture was heated at 100 °C for 3 h. The reaction mixture was cooled to room temperature. To the mixture 2N HCl (3 mL) was added and stirred for 10 min. The mixture was extracted with ethyl acetate (3 × 10 mL). The combined organic layer was washed successively with H_2_O (5 × 10 mL) and brine (50 mL), dried over Na_2_SO_4_, and concentrated under reduced pressure. The crude reaction mixture was purified by silica-gel CC to provide **9u** (37 mg, 86% yield) as a white solid (*R*_f_ = 0.6, petroleum ether/ethyl acetate = 1:1). ^1^H NMR (300 MHz, CDCl_3_) δ 7.31 (t, *J* = 8.1 Hz, 1H), 6.94 (d, *J* = 7.7 Hz, 1H), 6.80 (d, *J* = 8.3 Hz, 1H), 6.40 (d, *J* = 5.2 Hz, 1H), 5.44 (d, *J* = 5.1 Hz, 1H), 5.14 (dd, *J* = 9.0, 6.3 Hz, 1H), 3.78 (s, 3H), 3.24 (p, *J* = 6.9 Hz, 1H), 3.06 (dd, *J* = 17.3, 9.0 Hz, 1H), 2.39 (dd, *J* = 17.3, 6.3 Hz, 1H), 2.05–1.97 (m, 9H), 1.22–1.14 (m, 6H), 1.02 (d, *J* = 6.7 Hz, 3H). ^13^C NMR (75 MHz, CDCl_3_) δ 171.13, 170.25, 169.81, 158.63, 142.83, 134.85, 130.28, 129.30, 119.72, 119.47, 109.52, 79.33, 77.49, 77.06, 76.64, 71.34, 66.77, 55.48, 55.41, 38.51, 27.22, 22.13, 21.43, 21.24, 21.09, 20.88, 20.57. HRMS Calcd for C_24_H_30_O_7_ ([M + Na]^+^): 453.1884; Found: 453.1875.

*(3S,3aR,4S,5R)*-*5,6*-*dihydroxy*-*1*-*isopropyl*-*3a*-*methyl*-*3,3a,4,5*-*tetrahydro*-*2H*-*cyclopenta[a]naphthalene*-*3,4*-*diyl diacetate* (**9v**). To a solution of compound **9u** (43 mg, 0.1 mmol) in THF (1.5 mL), BBr_3_ (48 μL, 0.5 mmol) was added dropwise at −78 °C and stirred for 2 h. The reaction mixture was warm to room temperature slowly, quenched with ice water (5 mL), and then extracted with ethyl acetate (3 × 5 mL). The combined organic layer was washed successively with brine (10 mL), dried over Na_2_SO_4_, and concentrated under reduced pressure. The crude reaction mixture was purified by silica-gel CC to provide **9v** (28 mg, 75% yield) as a white solid (*R*_f_ = 0.2, petroleum ether/ethyl acetate = 2:1) [42]. ^1^H NMR (300 MHz, CDCl_3_) δ 7.22 (t, *J* = 7.9 Hz, 1H), 6.92–6.76 (m, 2H), 5.40–5.24 (m, 2H), 5.13 (dd, *J* = 9.0, 6.0 Hz, 1H), 3.24–2.98 (m, 2H), 2.40 (dd, *J* = 17.4, 6.0 Hz, 1H), 2.06 (d, *J* = 2.8 Hz, 6H), 1.23–1.15 (m, 6H), 1.00 (d, *J* = 6.8 Hz, 3H). ^13^C NMR (75 MHz, CDCl_3_) δ 171.99, 171.23, 157.12, 142.39, 133.15, 130.22, 129.24, 119.31, 118.35, 115.59, 79.70, 77.45, 77.03, 76.60, 73.69, 69.27, 55.49, 38.59, 27.11, 22.89, 21.45, 21.31, 21.21, 20.93. HRMS Calcd for C_21_H_26_O_6_ ([M + Na]^+^): 397.1622; Found: 397.1617.

*(3aS,3bR,4S,10bR)*-*6*-*isopropyl*-*10*-*methoxy*-*2,2,3b*-*trimethyl*-*3a,3b,5,10b*-*tetrahydro*-*4H*-*cyclopenta[3,4]**naphtho[1,2*-*d][1,3]**dioxol*-*4*-*ol* (**9w**). To a stirred solution of compound **9p** (46 mg, 0.15 mmol) and Pyridinium *p*-Toluenesulfonate acid (5 mg, 0.02 mmol) in CH_2_Cl_2_ (2 mL), 2,2-dimethoxypropane (40 mg, 0.38 mmol) was added at room temperature. Then the reaction mixture was stirred for 6 h at 40 °C. The solvent was removed under reduced pressure. The crude reaction mixture was purified by silica-gel CC to provide **9w** (51 mg, 67% yield) as a white solid (*R*_f_ = 0.3, petroleum ether/ethyl acetate = 3:1). ^1^H NMR (300 MHz, CDCl_3_) δ 7.27 (t, *J* = 8.0 Hz, 1H), 6.82 (t, *J* = 7.9 Hz, 2H), 5.59 (d, *J* = 6.8 Hz, 1H), 4.51 (d, *J* = 6.8 Hz, 1H), 4.24–4.14 (m, 1H), 3.88 (d, *J* = 1.4 Hz, 3H), 3.13–2.90 (m, 3H), 2.35 (dd, *J* = 16.9, 6.5 Hz, 1H), 1.44 (s, 3H), 1.24 (s, 3H), 1.16 (d, *J* = 6.9 Hz, 3H), 0.96 (d, *J* = 6.8 Hz, 3H), 0.91 (d, *J* = 1.4 Hz, 3H). ^13^C NMR (75 MHz, CDCl_3_) δ 158.42, 143.06, 135.70, 130.93, 128.97, 122.69, 119.73, 109.23, 109.01, 80.42, 79.79, 77.50, 77.08, 76.65, 71.51, 55.88, 55.22, 41.71, 27.08, 27.00, 25.50, 22.88, 21.84, 21.15. HRMS Calcd for C_21_H_28_O_4_ ([M + Na]^+^): 367.1880; Found: 367.1879.

### 4.2. Biological Activity Evaluation

#### 4.2.1. No Inhibition Assay

BV-2 cells were from Peking Union Medical College, Cell Bank (Beijing, China) and maintained in DMEM medium supplemented with 10% fetal bovine serum, penicillin (100 U/mL), and streptomycin (100 μg/mL) in a humidified incubator containing 95% air and 5% CO_2_ at 37 °C. Cell viability was measured by CCK8 assay. The NO production inhibitions were assessed as previously described [33]. BV-2 cells were seeded in 96-well plates at a concentration of 2 × 10^4^ cells/well. After 24 h of incubation at 37 °C, the cells were treated with 1 μg/mL of lipopolysaccharide (LPS) (*Escherichia coli* 0111:B4, Sigma-China, Shanghai, China) and various concentrations of test compounds (DMSO as solvent) for 24 h. An equal amount of DMSO and LPS served as the controls; quercetin (J&K Scientific, Beijing, China) was taken as the positive control. The NO production in microglial culture was measured indirectly from the supernatant using a NO assay kit (Beyotime Institute of Biotechnology Company, Jiangsu, China) based on the Griess reaction. Briefly, 50 μL of the culture supernatant was put into a new 96-well plate. Subsequently, 50 μL of Griess reagents I and II was added to each well. The absorbance at 540 nm was measured on a microplate reader (Mode 680, Bio-Rad, Tokyo, Japan). The concentration of nitrite was measured by using a standard curve via the standard sodium nitrite solutions. The IC_50_ was calculated as the concentration that reduced NO production by 50%.

#### 4.2.2. Neurite Outgrowth Assay

Cell culture. Rat pheochromocytoma PC12 cell line was purchased from the Shanghai Institute of Biochemistry and Cell Biology (Chinese Academy of Sciences), and maintained in DMEM containing 10% heat-inactivated HS and 5% FBS with 1% penicillin/streptomycin at 37 °C in a humidified 5% CO_2_ atmosphere. During logarithmic growth phase, PC-12 cells (2 × 10^4^ cells/well) were placed into a 24-well plate coated with poly-l-lysine (Sigma-Aldrich-China, Shanghai, China) in normal serum medium for 24 h, followed with low-serum (2% HS and 1% FBS) medium treatment for 14 h. The cells were treated with compounds at various concentrations (20 μm, DMSO as solvent) in the absence or presence of NGF (10 ng/mL). Cells treated with 10 ng/mL of NGF served as a positive control. The same concentration experiment was repeated in three wells. After an additional 72 h of incubation, neurite outgrowth of PC-12 cells was observed under an inverted microscope using phase-contrast objectives and photographed with a digital camera. Morphological analysis and quantification of neurite bearing cells were performed using phase-contrast microscope as described previously [33]. For each well, five images were randomly captured under a microscope. In each selected field, at least 100 cells were scored. The cells with neurites that were greater than or equal to the length of the cell body served as positive hits and then expressed as a percentage of the total cell number in the five fields.

#### 4.2.3. Cell Viability Assay

BV-2 cells were seeded into 96-well plates at 2 × 10^4^ cells/well. After 24 h, drugs were added and incubated for another 24 h. 10 μL CCK8 (APExBIO, Houston, TX, USA) was added to each well. After 4 h, the microplate reader measured absorbance at 450 nm.

#### 4.2.4. Neuroprotective Effects against H_2_O_2_-Induced Oxidative Injury in PC12 Cells

PC12 cells were maintained in DMEM supplemented with 10% heat-inactivated horse serum and 5% fetal bovine serum in humidified 5% CO_2_/95% air at 37 °C. All cells were cultured in collagen coated culture dishes or flasks. The medium was changed every other day. Before treatment, cells were plated at an appropriate density on culture plates or dishes according to each experimental scale and cultured for 24 h. Cells were pretreated for 6 h with various concentrations of samples. Then, the medium was refreshed without adding sample, and the cells were exposed to 400 µm H_2_O_2_ for another 16 h [34].

#### 4.2.5. Cytokine Release Was Measured Using an ELISA Kit

BV-2 microglial cells were cultured in a 24-well plate at 2 × 10^5^ cells/mL and 500 μL for 24 h. Then stimulated with the control group, model group, and administration group for 24 h. The supernatant was collected from the culture, and supernatant was used to measure the concentrations of TNF-α using an ELISA kit (Boster Biological Technology, Pleasanton, CA, USA) [35].

### 4.3. Molecular Docking Studies

Molecular docking simulations were performed using the software SYBYL-X 2.0 as the previous studies [45]. We performed a re-docking of the extracted ligand present in the iNOS complex. The similarity range were 0.3–0.6 (<0.8), which indicated the docking protocol was feasible. The three-dimensional (3D) crystal structure of iNOS (PDB code: 3E7G) was obtained from the RCSB Protein Data Bank.

## 5. Conclusions

In summary, we have enantioselectively synthesized for the first time hamigeran B norditerpenoid analogs with high yield. This methodology provides a route for the rapid assembly of the hamigeran pool in a practical fashion and sufficient quantities to facilitate drug discovery. In this work, the neurotrophic and anti-neuroinfammatory activities of hamigeran B analogs were first investigated. Cytotoxicity experiments, H_2_O_2_-induced oxidative injury assay, and ELISA reaction speculated that compounds may inhibit the TNF-α pathway to achieve anti-inflammatory effects on nerve cells. The SARs outlined provided an insight into the interactions between iNOS and a new class of the most potent NO inhibitors (**9c**, **9q**) to facilitate the further structural modification of this compound class, and two derivatives **9q** and **9o** have potential as dual-role therapeutic agents for AD treatment. Our investigations would expand and enrich the chemical and biological diversity of hamigeran-like norditerpenoid analogs.

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
