# Peer review of "Synthesis and Biological Evaluation of Diversified Hamigeran B Analogs as Neuroinflammatory Inhibitors and Neurite Outgrowth Stimulators"

_marinedrugs, 2020, doi:10.3390/md18060306_

Round 1

Reviewer 1 Report

The Authors describe an efficient synthesis of a series of new simplified hamigeran B and 1‑hydroxy-9-epi-hamigeran B norditerpenoid analogs, structurally related to cyathane diterpenoid scaffold, and their anti-neuroinflammatory and neurite outgrowth stimulating (neurotrophic) activity.

The paper explains clearly the synthesis of a list of marine-derived natural compounds followed by basic evaluation of anti-neuroinflammatory and neurotrophic activities limited to morphological analysis of neurite-outgrowth stimulation, and nitric oxide synthase inhibition as anti-neuroinflammatory activity.

The biological part of the study should have received a deeper understanding by better detailed testing on, for example, neurotrophin receptors and cell signaling pathways involvement, as well as changes of the expression of inflammatory cytokines and markers of neuronal differentiation. But this can probably be the issue for a different study that Authors would eventually afford in collaboration with expert pharmacologists or biochemists.

The Authors can at least briefly explain the method used to measure the inhibitory effects of the hamigeran derivatives on LPS-stimulated NO production in BV2 cells. This is helpful for the reader.

The investigation on the putative binding mode of the inhibitors on the inducible nitric oxide synthase (iNOS), carried out by molecular modeling gives an explanation/hypothesis of the mechanism of protection, but excludes the possible effects on the molecular events during peroxynitrite formation that require the presence of superoxide ions.

NO production by iNOS activation is a paradigm for inflammatory condition, but detrimental effects can also be attributable to “simply” oxidative stress. The Authors should at least mention and discuss this condition, or experimentally evaluate it. This could provide further properties.

The Authors could discuss the limits of their speculations implementing by a deeper description of the various inflammatory or neurotrophic scenarios.

Line 184: I’m probably wrong, but is it correct to say “binding binding-length” or it’s mistyped?

The Authors describe an efficient synthesis of a series of new simplified hamigeran B and 1‑hydroxy-9-epi-hamigeran B norditerpenoid analogs, structurally related to cyathane diterpenoid scaffold, and their anti-neuroinflammatory and neurite outgrowth stimulating (neurotrophic) activity.

The paper explains clearly the synthesis of a list of marine-derived natural compounds followed by basic evaluation of anti-neuroinflammatory and neurotrophic activities limited to morphological analysis of neurite-outgrowth stimulation, and nitric oxide synthase inhibition as anti-neuroinflammatory activity.

The biological part of the study should have received a deeper understanding by better detailed testing on, for example, neurotrophin receptors and cell signaling pathways involvement, as well as changes of the expression of inflammatory cytokines and markers of neuronal differentiation. But this can probably be the issue for a different study that Authors would eventually afford in collaboration with expert pharmacologists or biochemists.

The Authors can at least briefly explain the method used to measure the inhibitory effects of the hamigeran derivatives on LPS-stimulated NO production in BV2 cells. This is helpful for the reader.

The investigation on the putative binding mode of the inhibitors on the inducible nitric oxide synthase (iNOS), carried out by molecular modeling gives an explanation/hypothesis of the mechanism of protection, but excludes the possible effects on the molecular events during peroxynitrite formation that require the presence of superoxide ions.

NO production by iNOS activation is a paradigm for inflammatory condition, but detrimental effects can also be attributable to “simply” oxidative stress. The Authors should at least mention and discuss this condition, or experimentally evaluate it. This could provide further properties.

The Authors could discuss the limits of their speculations implementing by a deeper description of the various inflammatory or neurotrophic scenarios.

Line 184: I’m probably wrong, but is it correct to say “binding binding-length” or it’s mistyped?

Author Response

Dear reviewers:

Thank you for your letter and comments concerning our manuscript entitled " Synthesis and Biological Evaluation of Diversified Hamigeran B Analogs as Neuroinflammatory Inhibitors and Neurite Outgrowth Stimulators " (marinedrugs-810838). Your comments are valuable and very helpful for revising and improving our paper, as well as the important guiding significance to our researches. We have studied the comments carefully and made relevant corrections which we hope meet your approval. Revised portion in the manuscript are marked in red. The point to point responses to the comments are listed as follows:

  • Point 1: The biological part of the study should have received a deeper understanding by better detailed testing on, for example, neurotrophin receptors and cell signaling pathways involvement, as well as changes of the expression of inflammatory cytokines and markers of neuronal differentiation. But this can probably be the issue for a different study that Authors would eventually afford in collaboration with expert pharmacologists or biochemists.

Response: Thank you for your attention to our article and raising the above point. This suggestion is very good and coincides with our plan. Our next plan is neurotrophin receptors and cell signaling pathways involvement, and it's being studied.

  • Point 2: The Authors can at least briefly explain the method used to measure the inhibitory effects of the hamigeran derivatives on LPS-stimulated NO production in BV2 cells. This is helpful for the reader.

Response: The method for proinflammatory cytokine release was added in the “Anti-Inflammatory Assay” section of manuscript, which was marked in red.

“BV-2 cells were seeded on a 96 well plate (4 × 104 cells/well) and treated with 200 ng/ml LPS in the presence or absence of different concentrations of LNE for 24 h. the concentration of nitrite (NO2), a soluble oxidation product of NO, in the culture media was measured using Gries reagent (0.1%N-1-napthylethylenediamine dihydrochloride and 1% sulfanilamide in 5% phosphoric acid). Fifty microliters of supernatant was mixed with an equal volume of the Gries reagent and OD was measured at 570 nm. L-NMMA, a well-known nitric oxide synthase (NOS) inhibitor, was used as a positive control. Sodium nitrite was used as a standard to calculate NO2 concentrations.”

  • Point 3: The investigation on the putative binding mode of the inhibitors on the inducible nitric oxide synthase (iNOS), carried out by molecular modeling gives an explanation/hypothesis of the mechanism of protection, but excludes the possible effects on the molecular events during peroxynitrite formation that require the presence of superoxide ions.NO production by iNOS activation is a paradigm for inflammatory condition, but detrimental effects can also be attributable to “simply” oxidative stress. The Authors should at least mention and discuss this condition, or experimentally evaluate it. This could provide further properties.

Response: Due to time pressure, we have explained the mechanism of anti-inflammation and added the peroxynitrite case in the inflammatory occurrence, which is marked in red in manuscript.

“The overproduction of NO is an important signal in the inflammatory occurrence together with other pro-inflammatory stimuli. When excessively produced, NO, as a reactive radical, directly damages functional normal tissues. Moreover, it can also react with superoxide anion radical to form the even stronger oxidant peroxynitrite (ONOO). In this article we discuss the way that NO radical directly damages functional normal tissues.”

  • Point 4: The Authors could discuss the limits of their speculations implementing by a deeper description of the various inflammatory or neurotrophic scenarios.

Response: We have added the description in the “Discussion” section, which is marked in red in manuscript. “Our investigation carried out by molecular modeling gives an explanation of the mechanism of protection. The hypothesis is based on the inhibitors on the inducible nitric oxide synthase (iNOS), excludes the effects on the molecular events during peroxynitrite formation.”

  • Point 5: Line 184: I’m probably wrong, but is it correct to say “binding binding-length” or it’s mistyped?

Response: Yes, it’s mistyped. The correct is “binding-length”.

Reviewer 2 Report

Li et al. report that they have synthesized new hamigeran B analogues that display either anti-inflammatory and/or neurite outgrowth stimulating properties. The authors propose that these new molecules may function to reduce neuroinflammation and stimulate the production of new neurites. Conditions associated with the overproduction of NO include septic shock, inflammation and neurodegenerative disorders and therefore the newly synthesized compounds could potentially be of high value if they can indeed block NO production without toxic effects.

The manuscript describes the synthesis and properties of the new molecules in considerable detail but suffers from a very succinct overview of the biological properties.

Methods

  1. Please indicate the origin (Company, City, Country) of all the reagents.
  2. LPS is not even mentioned. Indicate the exact type of LPS used, concentration and length of the assay.
  3. The ‘anti-inflammatory assay’ should be renamed ‘iNOS induction assay’ – or something similar – to be more specific.
  4. The ‘anti-inflammatory assay’ and the ‘neurite outgrowth assay’ papers are referenced and indicated “as previously described”. Oddly, in the Results section (for both techniques) there are papers referenced which are different from the ones cited in the methods, this is confusing. For both techniques, even though the reader is directed to other articles, there should be at least a brief description of the method used and any deviation from the reference articles should be clearly indicated.

Results

The authors suggest that these new molecules are potential therapeutic candidates that have the advantage over endogenous neurotrophins of not having “protein properties* and high molecular weight” (line 48) for the treatment of neurodegenerative diseases, thereby suggesting that they have discovered compounds that can effectively replace neurotrophins. In Figure 3, however, the neurons are exposed to low (10ng/mL) concentrations of NFG along with the compounds. Do the new molecules, when added alone, have any effect on neurite outgrowth? If not, re-write the statement on line 48 so as to not give the impression that these can replace neurotrophins. The observation that some of the compounds seem to enhance the activity of neurotrophins should certainly be discussed however.

*Please indicate what is meant by ‘protein properties’.

It’s unclear why 20 mM was chosen as the concentration for the test compounds in the neurite outgrowth assay. If EC50 data is available for at least the lead compounds, please include it.

Figure 3 (top): Add arrowheads to show which neurites would be counted. Add scale bar.

It’s not appropriate to make a direct link between NO production and inflammation. In Figure 4, the Y-axis should therefore be labeled NO and should ideally show the actual levels in µM per mL rather than a percentage (we have to assume that this is relative to untreated BV-2 cells but this is not specified). The legend should be more descriptive, and the statistical parameters should be indicated, including ‘n’ value, what the error bars represent. The statistical test used to assign significance should be indicated and the p-value should also be noted. In Figure 3 the statistically significant compounds are represented by a purple bar but in Figure 4 meaning of the purple bars is not indicated (indicated in the text but should be noted in the legend, and I assume that it’s a mistake that the 9f bar is not purple). In the text, IC50 values for some of the compounds are provided. Please provide the concentration curves that allowed these values to be determined (at least in the supplementary data).

            The effect on NO production could technically be due to toxicity on the cells. At least for the lead compounds viability (live/dead) assay and a toxicity assay such as a lactate dehydrogenase release assay should be performed. Another possible cause of a reduction of NO would be if the compounds have an effect on the expression of iNOS, rather than a direct active-site blockade. The expression of iNOS should therefore be evaluated in response to the compounds (either at the protein level by Western blot or at least at the mRNA level by RT-PCR).

            If the authors wish to make a link to inflammation, it is necessary to evaluate the effect of the compounds on the levels of pro-inflammatory cytokines such as TNF-alpha and IL-1beta ideally at the protein level, but at least at the mRNA level.

Discussion

The discussion is very short and includes material that I would consider to be results, ie. the molecular modeling (Figure 6). A comparison with the existing pharmacopeia for the NO blockade and the stimulation of neurite outgrowth would be useful. While a mechanism for the blockade of NO production is proposed, there is no proposed mechanism for the promotion of neurite outgrowth. An analysis of the limits of the current work and proposed next steps would also be useful.

Statistics

Figure 3: What test is used? Are the error bars standard deviation or standard error of the mean? What is the ‘n’ value (of completely separate experiments)? How many cells were counted?

Line 37 Delete ‘s’ for ‘dementia’

Line 38 Suggestion: ‘any available’ after ‘without’

Line 44 Insert ‘for’ after ‘search’

Line 48 Replace ‘disable’ by ‘disqualified’ or ‘eliminated’

Line 49 Replace ‘to’ by ‘for’

Line 143 The last part of this sentence should be re-written.

Author Response

Dear reviewers:

Thank you for your letter and comments concerning our manuscript entitled " Synthesis and Biological Evaluation of Diversified Hamigeran B Analogs as Neuroinflammatory Inhibitors and Neurite Outgrowth Stimulators " (marinedrugs-810838). Your comments are valuable and very helpful for revising and improving our paper, as well as the important guiding significance to our researches. We have studied the comments carefully and made relevant corrections which we hope meet your approval. Revised portion in the manuscript are marked in red. The point to point responses to the comments are listed as follows:

  • Point 1: The manuscript describes the synthesis and properties of the new molecules in considerable detail but suffers from a very succinct overview of the biological properties.

Response: Thank you for your attention to our article and raising the above point. The biology section has been refined through your suggestion.

Methods:

  • Point 2: Please indicate the origin (Company, City, Country) of all the reagents. LPS is not even mentioned. Indicate the exact type of LPS used, concentration and length of the assay.

Response: According to your suggestion, we have added the detail origin (Company, City, Country) of the reagents in the section of ‘Materials and Methods’.

  • Point 3: The ‘anti-inflammatory assay’ should be renamed ‘iNOS induction assay’ – or something similar – to be more specific.

Response: We have modified the ‘anti-inflammatory assay’ into the ‘NO Inhibition Assay’.

  • Point 4: The ‘anti-inflammatory assay’ and the ‘neurite outgrowth assay’ papers are referenced and indicated “as previously described”. Oddly, in the Results section (for both techniques) there are papers referenced which are different from the ones cited in the methods, this is confusing. For both techniques, even though the reader is directed to other articles, there should be at least a brief description of the method used and any deviation from the reference articles should be clearly indicated.

Response: This suggestion is very useful for improving our paper. We have added the detail description that used to measure the biological activity in the section of ‘Materials and Methods’.

Results:

  • Point 5: The authors suggest that these new molecules are potential therapeutic candidates that have the advantage over endogenous neurotrophins of not having “protein properties* and high molecular weight” (line 48) for the treatment of neurodegenerative diseases, thereby suggesting that they have discovered compounds that can effectively replace neurotrophins. In Figure 3, however, the neurons are exposed to low (10ng/mL) concentrations of NFG along with the compounds. Do the new molecules, when added alone, have any effect on neurite outgrowth? If not, re-write the statement on line 48 so as to not give the impression that these can replace neurotrophins. The observation that some of the compounds seem to enhance the activity of neurotrophins should certainly be discussed however. *Please indicate what is meant by ‘protein properties’.

Response: The compounds displayed nerve growth factorinduced neurite outgrowth-promoting activity. The protein properties mean the complex structures and high molecular weight. We have re-write the statement on line 48-51. ‘However, their protein properties such as complex structures and high molecular weight eliminated them for using as clinical medication. Hence, the search for small molecules with the NGF biological functions or the NGF-induced neurotrophic activity arouses high interest.’

  • Point 6: It’s unclear why 20 mM was chosen as the concentration for the test compounds in the neurite outgrowth assay. If EC50 data is available for at least the lead compounds, please include it.

Response: The reason of chosing concentration 20 μM was according to our previous work. In order to highlight the compounds’ effects, we reduced the NGF to 10 ng/mL. When the concentration is greater than 20 μM, that does not have much significance in the application of drugs.

Yin X., Wei J., Wang W. W., Gao Y. Q., Stadler M., Kou, R. W., Gao J. M. New cyathane diterpenoids with neurotrophic and anti-neuroinflammatory activity from the bird's nest fungus Cyathusa fricanus. Fitoterapia., 2019, 134, 201–209.

  • Point 7: Figure 3 (top): Add arrowheads to show which neurites would be counted. Add scale bar.

Response: We have added arrowheads and scale bar in Figure 3 (top).

  • Point 8: It’s not appropriate to make a direct link between NO production and inflammation. In Figure 4, the Y-axis should therefore be labeled NO and should ideally show the actual levels in µM per mL rather than a percentage (we have to assume that this is relative to untreated BV-2 cells but this is not specified). The legend should be more descriptive, and the statistical parameters should be indicated, including ‘n’ value, what the error bars represent. The statistical test used to assign significance should be indicated and the p-value should also be noted. In Figure 3 the statistically significant compounds are represented by a purple bar but in Figure 4 meaning of the purple bars is not indicated (indicated in the text but should be noted in the legend, and I assume that it’s a mistake that the 9f bar is not purple). In the text, IC50 values for some of the compounds are provided. Please provide the concentration curves that allowed these values to be determined (at least in the supplementary data).

Response: The Y-axis is the NO inhibition, we have corrected. We have checked the Figure 3 and Figure 4 carefully, added ‘n’, error bars, marked the purple bars’ meaning. We have filled the same color in each bar of Figure 4. We have added the IC50 concentration curves in the supplementary data.

  • Point 9: The effect on NO production could technically be due to toxicity on the cells. At least for the lead compounds viability (live/dead) assay and a toxicity assay such as a lactate dehydrogenase release assay should be performed. Another possible cause of a reduction of NO would be if the compounds have an effect on the expression of iNOS, rather than a direct active-site blockade. The expression of iNOS should therefore be evaluated in response to the compounds (either at the protein level by Western blot or at least at the mRNA level by RT-PCR).

Response: Thanks for the constructive comments of the reviewer. This suggestion is very useful and important for us. Based on reviewers’ suggestions, recently we have supplemented the cytotoxicity experiment,and found that none of the compounds had significant cytotoxic activity. The relevant experimental datas have been added to thentext.

  • Point 10: If the authors wish to make a link to inflammation, it is necessary to evaluate the effect of the compounds on the levels of pro-inflammatory cytokines such as TNF-alpha and IL-1beta ideally at the protein level, but at least at the mRNA level.

Response: Thanks for the constructive comments of the reviewer. Regarding the issues mentioned, the author needs further research. Therefore, the author used ELISA experiments to detect the secretion of TNF-α, and the corresponding test results have been added to the text. Due to time, the ELISA test of IL-β has not been carried out. The specific results of the in-depth study will be carried out in future studies.

Discussion

  • Point 11: The discussion is very short and includes material that I would consider to be results, ie. the molecular modeling (Figure 6). A comparison with the existing pharmacopeia for the NO blockade and the stimulation of neurite outgrowth would be useful. While a mechanism for the blockade of NO production is proposed, there is no proposed mechanism for the promotion of neurite outgrowth. An analysis of the limits of the current work and proposed next steps would also be useful.

Response: The discussion section including the structure-activity relationships and molecular modeling study. Through the analysis of the structure-activity relationship and molecular modeling study, the activity results are discussed and mutually supported. The study of neurotrophic activity is in the initial stage, and it was compared with NGF. The reasonable mechanism for the promotion of neurite outgrowth has not been put forward, this and comparison with the existing pharmacopeia will be consummated in our next study.

Statistics

  • Point 11: Figure 3: What test is used? Are the error bars standard deviation or standard error of the mean? What is the ‘n’ value (of completely separate experiments)? How many cells were counted?

Response: For each well, 5 images were randomly captured under a microscope. In each selected field, at least 100 cells were scored. The cells with neurites that were greater than or equal to the length of the cell body served as positive hits and then expressed as a percentage of the total cell number in 5 fields. The error bars mean standard error and was marked in red below the Figure 3 and Figure 4. The value of ‘n’ is 3. Every image has100-200 cells.

  • Point 12: Line 37 Delete ‘s’ for ‘dementia’

Response: We have deleted ‘s’.

  • Point 13: Line 38 Suggestion: ‘any available’ after ‘without’

Response: We have insert ‘any available’ after ‘without’.

  • Point 14: Line 44 Insert ‘for’ after ‘search’

Response: We have insert ‘for’ after ‘search’.

  • Point 15: Line 48 Replace ‘disable’ by ‘disqualified’ or ‘eliminated’

Response: We have replaced ‘disable’ by ‘eliminated’ in line 49.

  • Point 16: Line 49 Replace ‘to’ by ‘for’

Response: We have replaced ‘to’ by ‘for’.

  • Point 17: Line 143 The last part of this sentence should be re-written.

Response: We have re-written last part of this sentence as ‘as the less deliquescent than quercetin’.

Reviewer 3 Report

The authors present the synthesis of Hamigeran B analogues and their biological activity in neurite growth stimulation and for nitric oxide inhibition. The research work does has merit but it can be improved on several fronts.

I recommend publication after authors have incorporated the following corrections.

The activity for neurite growth stimulation reported is very moderate and therefore should not be over emphasized. The nitric oxide inhibition is in single digit micro molar range for the best compounds which is a good starting point for further improvement.

Since the main focus of the study is synthesis of analogues of Hamigeran B, all analogues synthesized should be clearly characterized which is not the case.

There is no synthetic scheme provided for all the analogues prepared. I strongly suggest the authors to provide this in the main text for the reader’s benefit.

There is no characterization provided for compound 7.

No synthetic details provided for compound 11.

No proton NMR provided for compound 16.

Any intermediate prepared should have at least two characterizations. For compounds which has only proton NMR please provide at least least count mass spectra.

There are many compounds for which the proton NMR does not match the number reported in the experimental. Please refer to the attachment to correct the values.

Please go through the reference to make all reference uniform. 5, 7, 22, 27, needs to be looked at.

Please add the following reference as it relates to first total synthesis of Hamigerans

Total Synthesis of Hamigerans and Analogues Thereof. Photochemical Generation and Diels−Alder Trapping of Hydroxy-o-quinodimethanes K. C. Nicolaou, David L. F. Gray, Jinsung Tae

https://pubs.acs.org/doi/10.1021/ja030498f

Author Response

Dear reviewers:

Thank you for your letter and comments concerning our manuscript entitled " Synthesis and Biological Evaluation of Diversified Hamigeran B Analogs as Neuroinflammatory Inhibitors and Neurite Outgrowth Stimulators " (marinedrugs-810838). Your comments are valuable and very helpful for revising and improving our paper, as well as the important guiding significance to our researches. We have studied the comments carefully and made relevant corrections which we hope meet your approval. Revised portion in the manuscript are marked in red. The point to point responses to the comments are listed as follows:

  • Point 1: The activity for neurite growth stimulation reported is very moderate and therefore should not be over emphasized. The nitric oxide inhibition is in single digit micro molar range for the best compounds which is a good starting point for further improvement.

Response: Thank you for your attention to our article and raising the above point. The activity for neurite growth stimulation reported is very moderate, hence we focus on the structure-activity relationship and molecular modeling study with anti-inflammatory activity emphatically in the section of discussion.

  • Point 2: Since the main focus of the study is synthesis of analogues of Hamigeran B, all analogues synthesized should be clearly characterized which is not the case. There is no synthetic scheme provided for all the analogues prepared. I strongly suggest the authors to provide this in the main text for the reader’s benefit.

Response: Thank you for your proposals in point. We are trying to design synthetic scheme, the result is not satisfied unfortunately. Because of the synthesis of the analogues is diverse, using synthetic scheme to describe is disorganized. The brief cover of analogues synthesized was added in the manuscript, that was marked in red.

“Protected the hydroxyl of 9a afforded the corresponding compound 9h-n. Hydrogenation of 9h and 9g proceeded very efficiently to provide the hydrogenated product 9c and 9e. The compounds 9h, 9m and 9n were constructed via esterification from 9a. Hydrolysis and concomitant oxidation proceeded of 9c to produce 9d, followed condensation afforded 9g. Using trifluoroacetic acid to treat compound 9e afforded the product 9f. Compounds 9q and 9s were constructed via removing methyl on the phenolic group of 9o and 9r. The compound 9t was established through eliminated 1-Hydroxy of 9s simply. Protected the hydroxyl of 9p afforded the corresponding compound 9u-w.”

  • Point 3: There is no characterization provided for compound 7.

Response: Because of the aldehyde 7 is not stable, we are failed to obtain the pure spectrogram. Based on our previous work as well as the reported literature, once get the crude aldehyde 7, we are straight to the next step.

Our previous literature: 1. Total synthesis of cyrneines A-B and glaucopine C. Nat. Commun. 2018, 9, 2148. https://www.nature.com/articles/s41467-018-04480-6.

  1. A Total Synthesis of (-)-Hamigeran B and (-)-4-Bromohamigeran B. Org. Lett. 2018, 20, 3687–3690. https://pubs.acs.org/doi/10.1021/acs.orglett.8b01490.
  • Point 4: No synthetic details provided for compound 11. No proton NMR provided for compound 16. Any intermediate prepared should have at least two characterizations. For compounds which has only proton NMR please provide at least least count mass spectra.

Response: The compound 11 was provided as the previous study and in the manuscript are marked in red. We are failed to obtain the pure NMR, the by-product has the similar polarity. Nonetheless, we got the HRMS and next step the by-product can be removed.

Our previous literature: Total synthesis of cyrneines A-B and glaucopine C. Nat. Commun. 2018, 9, 2148. https://www.nature.com/articles/s41467-018-04480-6.

  • Point 5: There are many compounds for which the proton NMR does not match the number reported in the experimental. Please refer to the attachment to correct the values.

Response: In order to confirm that the dates are correct, we have checked all the proton NMR carefully and made relevant corrections. The proton NMR of compounds 9a-i, 9k-o and 9q were modified. The active hydrogen in compounds 8, 10, 17, 9p, 9s and 9v have no peak.

  • Point 6: Please go through the reference to make all reference uniform. 5, 7, 22, 27, needs to be looked at.

Response: To make sure all the reference uniform, we have studied the comments carefully and made relevant corrections.

  • Point 7: Please add the following reference as it relates to first total synthesis of Hamigerans Total Synthesis of Hamigerans and Analogues Thereof. Photochemical Generation and Diels−Alder Trapping of Hydroxy-o-quinodimethanes C. Nicolaou, David L. F. Gray, Jinsung Tae https://pubs.acs.org/doi/10.1021/ja030498f

Response: The literature above has been cited in this manuscript, referrence 27.

Round 2

Reviewer 2 Report

The authors have responded adequately to my previous concerns. I only have a comment regarding one the new results (Line 175, Fig 6): Please confirm that none of these treatments are significantly (ie. all p > 0.05) different from the untreated cells. The SD bars are very small, so I find it surprising that the difference - even if it is relatively small in all treatments compared to untreated - not to be significant in any of these experiments. If the difference is indeed small, yet significant, it should be indicated that 'while the compounds display mild toxicity, the toxicity cannot account for the other observed effects'.

Author Response

Dear reviewers:

Thank you for your letter and comments concerning our manuscript entitled " Synthesis and Biological Evaluation of Diversified Hamigeran B Analogs as Neuroinflammatory Inhibitors and Neurite Outgrowth Stimulators " (marinedrugs-810838). Your comments are valuable and very helpful for revising and improving our paper, as well as the important guiding significance to our researches. We have studied the comments carefully and made relevant corrections which we hope meet your approval. Revised portion in the manuscript are marked in red. The point to point responses to the comments are listed as follows:

Point 1: The authors have responded adequately to my previous concerns. I only have a comment regarding one the new results (Line 175, Fig 6): Please confirm that none of these treatments are significantly (ie. all p > 0.05) different from the untreated cells. The SD bars are very small, so I find it surprising that the difference - even if it is relatively small in all treatments compared to untreated - not to be significant in any of these experiments. If the difference is indeed small, yet significant, it should be indicated that 'while the compounds display mild toxicity, the toxicity cannot account for the other observed effects'.

Response: Thank you for your attention to our article and raising the above point. We have added the sentence ‘while the compounds display mild toxicity, the toxicity cannot account for the other observed effects’ in Line 174.

Reviewer 3 Report

The authors have made considerable improvement to the manuscript. These are some of the points to be addressed before the manuscript is accepted for publication.

In the introduction please mention neurite growth stimulation as moderate and not significant since the data suggests only moderate effect.

Figure two boxed structures in box two needs a double bond instead of dashed line in the cyclopentyl ring as all analogues have a double bond.

As mentioned before all compounds should have at least proton NMR and a low resolution mass spectra. Please see the attached pdf where the authors have to provide low resolution mass spectra.

The high resolution mass spectra for some of the compounds is shown as sodium salt but the calculated formula does not include the sodium salt. Please confirm if the mass obtained includes sodium and if so include them in the calculated mass.

If aldehyde 7 is unstable then this should be clearly mentioned in the text and given a reference

It is not clear from author's response as to why they didn't provide characterization data for 11 and 16. Please explain the reasons clearly in the text.

In the supplementary document the compound structures in the proton NMR spectra are missing numbers. Please add the numbers for reader's benefit.

Author Response

Dear reviewers:

Thank you for your letter and comments concerning our manuscript entitled " Synthesis and Biological Evaluation of Diversified Hamigeran B Analogs as Neuroinflammatory Inhibitors and Neurite Outgrowth Stimulators " (marinedrugs-810838). Your comments are valuable and very helpful for revising and improving our paper, as well as the important guiding significance to our researches. We have studied the comments carefully and made relevant corrections which we hope meet your approval. Revised portion in the manuscript are marked in red. The point to point responses to the comments are listed as follows:

  • Point 1: The authors have made considerable improvement to the manuscript. These are some of the points to be addressed before the manuscript is accepted for publication. In the introduction please mention neurite growth stimulation as moderate and not significant since the data suggests only moderate effect.

Response: Thank you for your attention to our article and raising the above point. We have changed ‘significant’ to ‘moderate’ in Line 18.

  • Point 2: Figure two boxed structures in box two needs a double bond instead of dashed line in the cyclopentyl ring as all analogues have a double bond.

Response: According to your suggestion, we have modified the dashed line in box two of Figure 2.

  • Point 3: As mentioned before all compounds should have at least proton NMR and a low resolution mass spectra. Please see the attached pdf where the authors have to provide low resolution mass spectra.

Response: This suggestion is very useful for improving our paper. We have added the high resolution mass spectras for compounds S1, 8, 10, 12,13, 14 and 17.

  • Point 4: The high resolution mass spectra for some of the compounds is shown as sodium salt but the calculated formula does not include the sodium salt. Please confirm if the mass obtained includes sodium and if so include them in the calculated mass.

Response: We have calculated formula include the sodium salt, and marked it in parentheses. In my opinion, it's more rigorous to describe the high resolution mass spectra.

  • Point 5: If aldehyde 7 is unstable then this should be clearly mentioned in the text and given a reference.

Response: We have mentioned the unstable property and cited the reference [26] in Lines 86 and 107, which is same to compound 16.

[26] Cao B. C., Wu G. J., Yu F., He Y. P., Han F. S. A Total Synthesis of (-)-Hamigeran B and (-)-4-Bromohamigeran B. Org. Lett. 2018, 20, 3687–3690.

  • Point 6: It is not clear from author's response as to why they didn't provide characterization data for 11 and 16. Please explain the reasons clearly in the text.

Response: We have explained the reasons clearly in Line 99-100 and109-110. The compound 11 was provided as our previous study, that are known compound and was identical to the references. We are failed to obtain the pure 16 as the by-product has the similar polarity. To our delight, the by-product can be removed next step.

  • Point 7: In the supplementary document the compound structures in the proton NMR spectra are missing numbers. Please add the numbers for reader's benefit.

Response: We have added numbers in the supplementary document.